# RVAS: Referring Video Active Exploration and Segmentation

**Hengrui Hu** [1]  **Weiwei Gao** [2]  **Zipei Zhang** [1]  **Henghui Ding** [1]

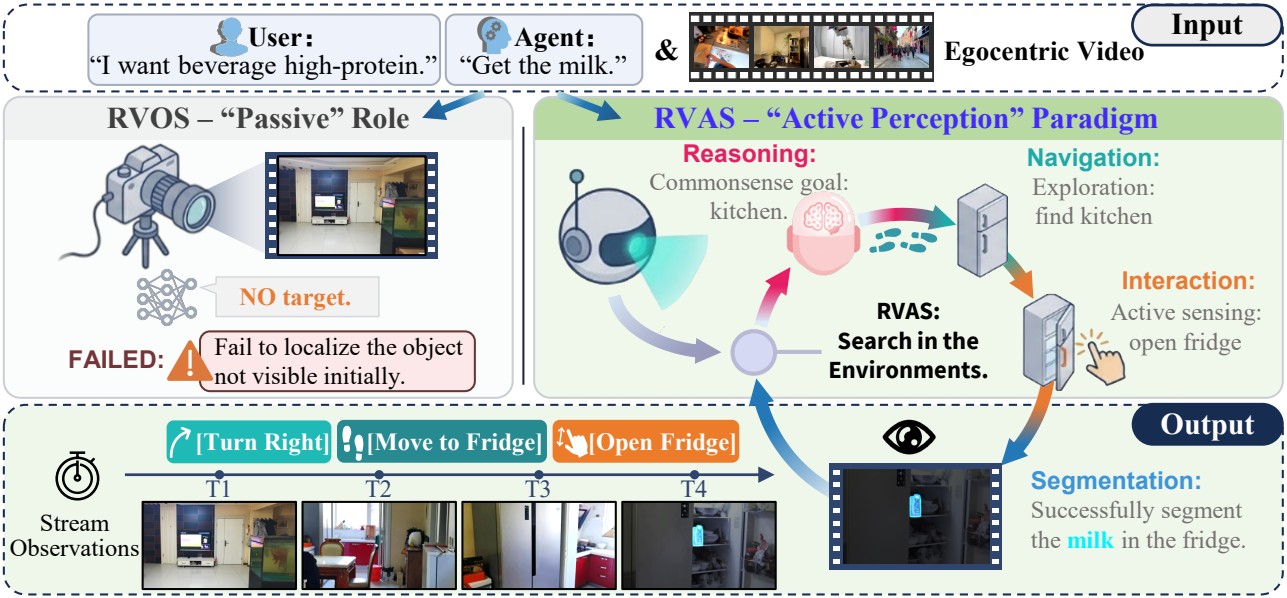

*Figure 1.* We introduce a new dataset and baseline for Referring Video Active Exploration and Segmentation (RVAS) task. RVAS aims to act like an active agent, rather than a passive perception model, that focuses on online planning and segmentation given streaming visual inputs and various referring expressions. RVAS requires the joint reasoning about commonsense and global goal, and exploration-oriented planning via open-set actions across navigation and interaction. This work provides a testbed for advancing comprehensive video model.

## Abstract

Existing referring video object segmentation (RVOS) is largely built on passive perception and assumes the target is already visible in the observed video, which limits real-world use when queries refer to objects beyond the current view. To address this gap, we introduce Referring Video Active Exploration and Segmentation (RVAS), a new task that focuses on reasoning about exploration policy and then locating and segmenting the object according to an input referring expression. To support RVAS, we build a large-scale dataset with manually annotated exploration actions and reference reasoning traces, enabling supervised training and evaluation. We benchmark representative RVOS and related video understanding baselines and find that they struggle to perform active target search and incur substantial overhead when coupled with online decision making. Motivated by these challenges, we propose LESA, a baseline framework that introduces a state controller and hierarchical memory for efficient streaming processing and sparse MLLM reasoning. LESA substantially reduces inference cost while maintaining competitive planning quality, and consistently improves segmentation accuracy on the RVAS dataset. Code is available at https://github.com/FudanCVL/RVAS.

[1]Institute of Big Data, College of Computer Science and Artificial Intelligence, Fudan University, Shanghai, China [2]Shanghai University of Engineering Science, Shanghai, China. Correspondence to: Henghui Ding <henghui.ding@gmail.com>.

*Proceedings of the 43rd International Conference on Machine Learning*, Seoul, South Korea. PMLR 306, 2026. Copyright 2026 by the author(s).

## 1. Introduction

Referring Video Object Segmentation (RVOS) (Seonguk, 2020; Ding et al., 2023) is an emerging field that aims to segment target objects indicated by language expressions,

requiring understanding of unconstrained instructions and fine-grained visual details. However, existing RVOS formulations are inherently passive. They assume that the full video is already observed and predict masks under fixed observations. Consequently, current models struggle when the referred object is occluded or out of view, limiting RVOS to offline applications such as video editing and limiting its deployment in applications that require active perception, despite its natural fit for human–machine interaction.

To bridge perception and actuation, Vision Object Navigation (VON) (Anderson et al., 2018; Majumdar et al., 2022) studies closed-loop policies that move toward a target, but most methods assume structured goal descriptions and restrict actions to locomotion. They also often rely on heavy sensors such as LiDAR or depth, or on full environment scans, which biases training toward simulation and limits robustness and scalability in real-world deployment. This raises a key question: Can we achieve goal-driven active exploration and precise segmentation in open-world environments using only low-cost monocular vision?

The open world setting means open vocabulary actions and unconstrained expressions. We introduce **R**eferring **V**ideo **A**ctive Exploration and **S**egmentation (**RVAS**), extending RVOS by requiring search-action planning before the target is observed, thereby better supporting applications in embodied AI and agent research. As shown in Figure 1, given user-provided expressions or high-level planning from agent, together with previously observed RGB frames, RVAS aims to generate intermediate textual action commands to explore the environment and maintain a mask prediction sequence, meanwhile in a streaming manner. These textual commands support open-set planning over both navigation and interaction, allowing for fully free decisions for actively capturing necessary information from the environment.

To support research on RVAS, we build RVAS dataset, a large-scale egocentric video dataset with 911 videos from 5 scene types and 91.7K frames. To better reflect embodied and real-world usage, we provide 4,633 queries in 3 forms, namely *object*, *intent*, and *instruction*. The latter two introduce implicit goals that are challenging for RVOS-style methods and require deeper reasoning, for example, "*I am thirsty*" may refer to beverages, and "*Check the current time*" may refer to a clock or a phone. In addition, RVAS includes 18.6K manually annotated exploration actions and step-wise human reasoning traces totaling 1,275K words, enabling supervised training for active exploratory planning. RVAS is evaluated on recorded egocentric video streams, providing a reproducible proxy evaluation for goal-conditioned exploration and segmentation under streaming observations.

Based on the proposed dataset, we benchmark representative methods under RVAS setting, covering RVOS models (Yan et al., 2024; Liang et al., 2025; Wei et al., 2025;

Yuan et al., 2025; Liu et al., 2025), generalist Multimodal LLMs (MLLMs) (Chen et al., 2024b), and streaming video methods (Chen et al., 2024a; Qian et al., 2025). The results reveal two significant challenges. First, RVAS tightly couples planning and segmentation in active exploration and must account for its dynamic, sequential decision-dependent process, where historical context dictates future attention, rather than treating exploration and segmentation as isolated sub-tasks. Second, RVAS highlights a streaming latency bottleneck. Many methods are too heavy for real-time use, as per-frame computation incurs prohibitive latency that conflicts with the streaming nature of RVAS.

To address these challenges, we draw inspiration from human cognition, continuously monitoring the environment with low-cost perception and invoking high-level reasoning only when necessitated by salient events. Guided by this principle, we propose the Language Environment-aware Segmentation Assistant (**LESA**), a framework that decouples continuous video monitoring from episodic high-level reasoning. Unlike existing methods that rely on passive, dense execution, LESA operates in a streaming event-triggered manner, maintaining an exploration-aware state for iterative planning and segmentation. To reduce streaming inference cost, LESA introduces a lightweight **State Controller** that processes streaming inputs and triggers the heavy MLLM backbone only at critical moments, meanwhile maintaining exploration-aware memories. To handle the tight coupling between planning and segmentation, we further design **Role-Structured Reasoning**, which employs a shared MLLM backbone to maintain a unified latent space and process mutually informed tasks by switching between Planner and Segmentor via task-specific tokens. LESA achieves about a $10\times$ planning speed-up over prior MLLM baselines with comparable reasoning accuracy, and demonstrates significant segmentation performance improvement. Overall, this work makes the following main contributions:

- **RVAS Task and Dataset.** We introduce Referring Video Active Exploration and Segmentation (RVAS) task, which extends passive perception to active, exploration-oriented planning in open-world environments, and build a large-scale egocentric video dataset with diverse expression types and high-quality reasoning traces.
- **Extensive Experiments and Analysis.** We conduct comprehensive experiments on RVAS, covering representative RVOS methods and related video understanding methods to reveal the core challenges of the task.
- **Efficient Streaming Framework.** We propose LESA, an efficient streaming perception, planning, and segmentation framework that couples exploration process and precise segmentation. LESA achieves significant speed-up and improved performance on RVAS.

**Conflict of Interest Disclosure.** The authors declare no

*Table 1.* Statistics of our **RVAS** and previous RVOS datasets. RVAS emphasizes efficient active exploration as well as concise segmentation, providing 18.6K steps of actions and 1,275K words of human-reasoning traces for training. In addition, RVAS introduces two new types of expressions, intent and instruction, posing new challenges of understanding implicit requirements and conducting exploratory planning.

| Dataset | Videos | Objects | Expressions | Multi Target | No Target | Intent Exp. | Instruct. Exp. | Actions | Explanatory Traces/words |
|---|---|---|---|---|---|---|---|---|---|
| DAVIS-RVOS | 90 | 90 | 205 | ✗ | ✗ | ✗ | ✗ | ✗ | ✗ |
| Ref-YoutubeVOS | 3,978 | 7,451 | 15,009 | ✗ | ✗ | ✗ | ✗ | ✗ | ✗ |
| MeViS | 2,006 | 8,171 | 28,570 | ✔ | ✗ | ✗ | ✗ | ✗ | ✗ |
| ReasonVOS | 91 | 458 | 458 | ✗ | ✗ | ✗ | ✗ | ✗ | ✗ |
| ReVOS | 1,042 | 9,084 | 35,074 | ✔ | ✔ | ✗ | ✗ | ✗ | ✗ |
| **RVAS (Ours)** | 911 | 1,468 | 4,633 | ✔ | ✔ | 1,458 | 1,197 | 18.6K | 1,275K |

financial conflicts of interest related to this work.

## 2. Related Works

### 2.1. Referring Video Object Segmentation

Referring Video Object Segmentation (RVOS) (Gavrilyuk et al., 2018; Seonguk, 2020; Ding et al., 2023; Ying et al., 2025a; Ding et al., 2025a) aims to segment objects throughout the video according to text expressions. Early methods (Liang et al., 2021; Botach et al., 2022; Luo et al., 2023; Yuan et al., 2024) primarily relied on deep networks to encode, align, and fuse cross-modal features. To enhance generalization capacity and realize reasoning segmentation (Yan et al., 2024; Bai et al., 2024), recent methods leveraged strong foundation models like GroundingDINO (Huang et al., 2025; Liang et al., 2025) and MLLMs (Wei et al., 2025; Lin et al., 2025; Yuan et al., 2025; Liu et al., 2025; Ying et al., 2024) for inference on sparse key frames, then apply SAM or SAM2 (Kirillov et al., 2023; Ravi et al., 2024) to propagate across the video. However, existing RVOS research is limited to passive perception and cannot decide how to further actively search for the targets based on the observed context. We extend the conventional RVOS setting to Referring Video Active Exploration and Segmentation (RVAS) to further explore this more challenging nature.

### 2.2. Vision Navigation

Vision Navigation focuses on locating a specified object in an unknown environment, which is fundamental to embodied systems. According to different prompt types, existing research can be categorized into vision object navigation (VON) (Chen et al., 2023; Cai et al., 2024; Yadav et al., 2023) and vision-language navigation (VLN) (Anderson et al., 2018; Mishra et al., 2023). Through end-to-end policy learning via RL and IL (Zheng et al., 2022; Ramrakhya et al., 2022) and spatial modeling of 2D map or 3D scene (Nanwani et al., 2023; Liu et al., 2024; Wang et al., 2024), previous approaches supported navigating according to object labels and language instructions. Most recent methods leveraged LLMs and VLMs (Yu et al., 2023; Nie

et al., 2025; Song et al., 2025) for long-horizon high-level reasoning, being compatible with more complex descriptions and instructions. However, real-world deployment of these methods remains limited by constrained action spaces and reliance on static, pre-scanned environments, which can be highly costly. To mitigate the constraints, we introduce RVAS, which adopts an open action space and runs on easily collected videos, providing a testbed for studying agent behavior in dynamic, interactive settings.

### 2.3. Action Anticipation

Action Anticipation (Ryoo, 2011; Tang et al., 2019; Sener et al., 2022) aims to forecast future actions by analyzing preceding actions. The action anticipation involves short-term anticipation (Li et al., 2018; Girase et al., 2023), which requests predicting immediate action, which is partially related to our work in form, and long-term anticipation (Zhang et al., 2024; Dalal et al., 2025), which forecasts a plausible action sequence over a long-horizon. In recent years, the related dominant paradigm has gradually shifted from transformer-based (Xu et al., 2021; Gong et al., 2022) toward utilizing MLLMs (Zhao et al., 2023; Pei et al., 2024; Cao et al., 2025; Wang et al., 2025). Several streaming video models (Chen et al., 2024a; Qian et al., 2025) further integrate online anticipation setting, as well. Despite formal similarity, action anticipation differs from our RVAS in core objectives. Anticipated action sequences are primarily required to be temporally and semantically plausible, while RVAS demands goal-oriented, biased planning that actively steers the agent toward a specified objective and rapidly adapts to updates in scene observation and understanding.

## 3. RVAS Dataset

### 3.1. RVAS Task Definition.

Given a language expression $\mathcal{T}$ and an initial visual observation $\mathcal{V}_0$, the goal of Referring Video Active Exploration and Segmentation (RVAS) is to iteratively generate navigation or interaction action commands $a_t$ to actively explore the unseen environment effectively. Once an updated observa-

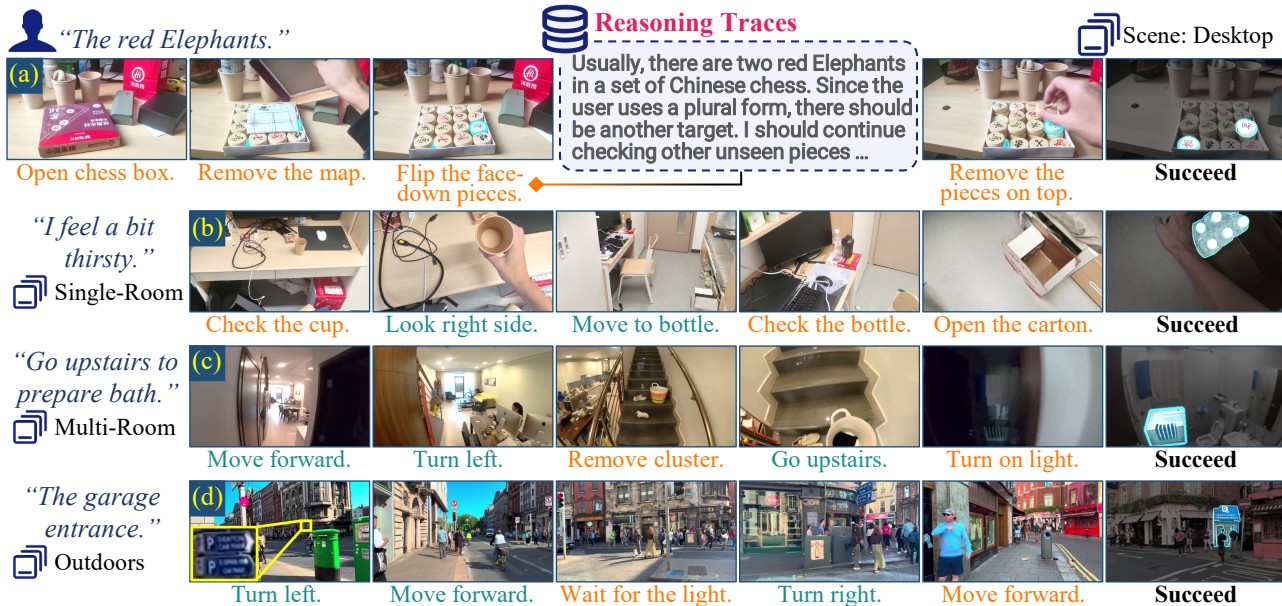

*Figure 2.* Examples from the **RVAS dataset**, with selected frames and action annotations. The actions involve **navigation** for locomotion, and fine-grained **interaction** with open-set objects, like "Remove clutter" or "Wait for the light", making them hard to be classified using a closed set of labels. Large-scale supported **reasoning traces** are provided, enabling the task training. **(a)** and **(b)** require commonsense reasoning and active exploration for reliable segmentation. **(c)** and **(d)** request planning based on user instructions and environment visual cues, demonstrating the complexity. RVAS also includes various expressions and scene types to make the dataset in-the-wild.

tion $\mathcal{V}_t$ is attained, the process continues until the target object referred to by $\mathcal{T}$ can be reliably localized at timestep $t_s$. Finally, a pixel-level segmented mask sequence $M_{0:t_s}$ is produced to precisely localize the object across both spatial and temporal dimensions.

### 3.2. Data Construction and Annotation.

To support the RVAS task, we construct a large-scale egocentric video dataset with referring expressions, ground-truth object masks, manually annotated exploration actions, and corresponding structured reasoning traces to enable training and evaluation aligned with human exploratory behavior, as shown in Figure 2. The videos focus on active exploration and search, and are collected from two sources: **(i)** publicly available egocentric videos and datasets, primarily Ego4D and EgoSchema (Grauman et al., 2022; Song et al., 2023; Mangalam et al., 2023), and **(ii)** self-recorded videos captured with participant consent, covering both scripted long-horizon exploration scenarios and free exploration in novel environments for naturalness.

Annotations are obtained through a multi-stage pipeline conducted by a group of trained annotators. First, annotators generate three types of language expressions, including object, intent, and instruction expressions, for selected target objects. A separate group further reviews the videos and expressions to remove unreasonable or ambiguous annotations and ensure correct object-expression correspondence.

Second, annotators identify decision points and annotate the corresponding exploration actions using a streaming annotation tool without access to future frames, while providing brief explanations for their decisions. Each candidate video is assigned to multiple annotators, and only samples with majority annotation agreement are retained to ensure natural and reasonable action sequences. Finally, full mask sequences and reference reasoning traces are annotated offline. Object masks are produced with assistance of the SAM2 tool, while reasoning traces are generated by jointly analyzing the given expression, historical frames, and decision explanations, following a structured `goal analysis`, `observation`, `inference`, and `planning` template. All annotations undergo quality control through annotator training with curated exemplars and subsequent audits.

### 3.3. Statistics and Analysis.

Table 1 summarizes key statistics of prior RVOS datasets (Khoreva et al., 2018; Seonguk, 2020; Bai et al., 2024; Yan et al., 2024) and the proposed RVAS dataset. RVAS contains 911 videos and 1,468 objects, annotated with 4,633 language expressions across three categories: 1,978 object, 1,458 intent, and 1,197 instruction expressions, introducing challenges in intent understanding and instruction following beyond object-centric referring, which is crucial in embodied systems. The videos span five scene types: desktop, single-room indoor, multi-room indoor, outdoor, and robotics scenarios, covering a broader range of real-

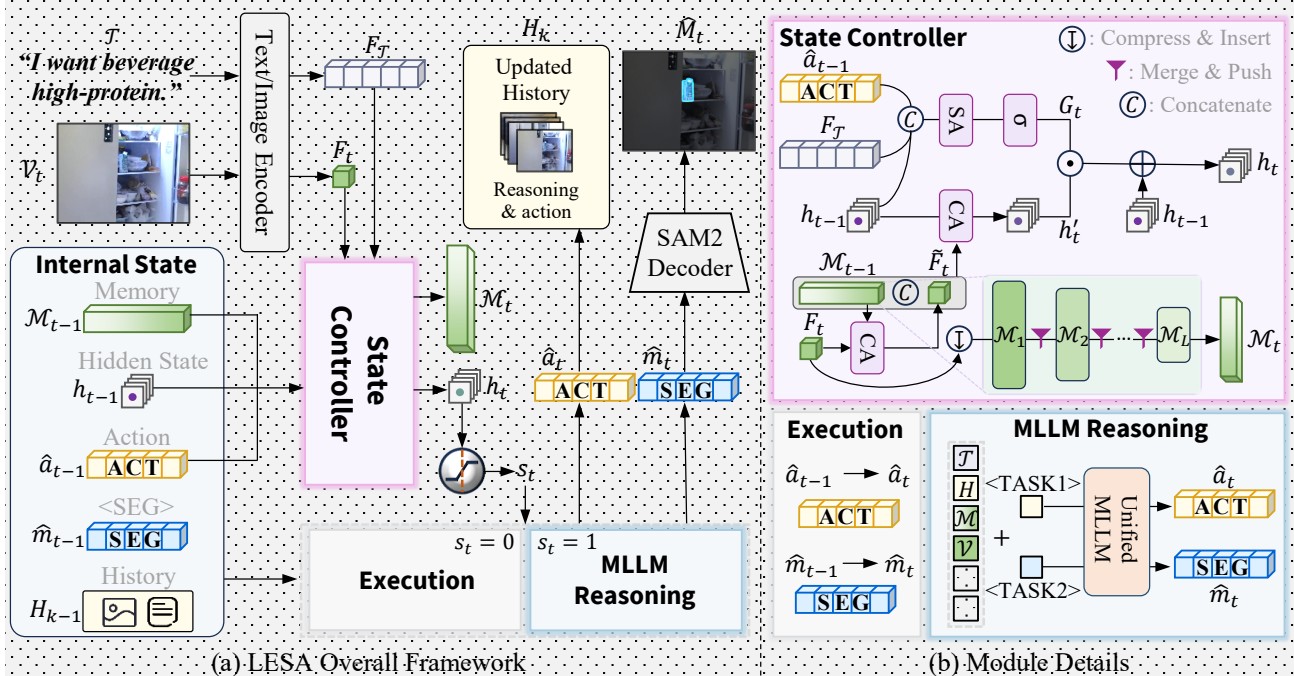

*Figure 3.* Overview of the proposed LESA framework. (a) The overall architecture, illustrating how the method maintains internal states via a state controller, triggers MLLM reasoning, and outputs actions and <SEG> tokens at each timestamp. (b) Detailed module mechanisms. The State Controller uses semantic-gated attention to update $h_t$ and $\mathcal{M}_t$ in an exploration-aware manner. Setting $s_t = 0$ denotes action execution (retaining the previous action and <SEG> token), while $s_t = 1$ triggers MLLM reasoning. Here, well-organized multimodal inputs, designed prompts (⊡), and specific <TASK> tokens activate a unified MLLM to generate $\hat{a}_t$ and $\hat{m}_t$, respectively.

world environments than most prior anticipation datasets focused on desktop and VON datasets centered on indoor-only settings, respectively. Figure 2 presents representative examples that require active exploration guided by goal-oriented reasoning, commonsense, language cues, and environmental observations, highlighting the necessity of principled exploration for a reliable segmentation. In addition, RVAS provides 18.6K manually annotated open-set actions and 1,275K words of high-quality human reasoning traces. These transparent reasoning traces, as shown in Figure 2 and abridged for clarity, demonstrate how reasonable actions are produced and enable supervision towards human-like active exploration. All videos are split into *train* and *val* sets with 611 and 300 videos, respectively, facilitating standardized training and evaluation for future RVAS research.

## 4. Methodology

**Overview.** Existing approaches adapted to RVAS typically rely on sparse frame sampling with dense MLLM inference for each new frame, which limits modeling of exploration dynamics while incurring high computational overhead. To address this, we propose the **L**anguage **E**nvironment-Aware **S**egmentation **A**ssistant (**LESA**) that intervenes only when necessary. Since action execution takes time and naturally follows a clip-wise structure, LESA first quickly scans the

new observation to determine whether to retain the previous plan. Upon action completion or detection of key cues, LESA invokes the role-structured MLLM backbone for high-level reasoning. By maintaining both key action histories and recent clip memories, LESA effectively models the exploration process for robust segmentation.

### 4.1. LESA Framework

Figure 3 illustrates the overall architecture of LESA. LESA consists of a MLLM backbone $\mathcal{F}_{\text{LLM}}$ for high-level episodic reasoning and a lightweight State Controller $\mathcal{F}_{\text{SC}}$ for online video state maintenance and trigger prediction. Given a streaming frame $\mathcal{V}_t$, the controller maintains a compact exploration-aware hidden state $h_t$ and a hierarchical visual memory $\mathcal{M}_t$, and predicts a binary control signal $s_t$:

$$h_t, \mathcal{M}_t, s_t = \mathcal{F}_{\text{SC}}(h_{t-1}, \mathcal{M}_{t-1}, \mathcal{V}_t). \quad (1)$$

The control signal $s_t \in \{0, 1\}$ determines whether the $\mathcal{F}_{\text{LLM}}$ should be invoked for a new round of reasoning:

$$(\hat{a}_t, \hat{m}_t) = \begin{cases} (\hat{a}_{t-1}, \hat{m}_{t-1}), & \text{if } s_t = 0, \\ \mathcal{F}_{\text{LLM}}(\mathcal{T}, \mathcal{V}_t, \{H_k\}, \mathcal{M}_t), & \text{if } s_t = 1. \end{cases} \quad (2)$$

When $s_t = 1$, $\mathcal{F}_{\text{LLM}}$ generates a new exploration action $\hat{a}_t$ and a <SEG> token $\hat{m}_t$, marking the start of a new exploration clip. The reasoning is conditioned on the referring

expression $\mathcal{T}$, the current frame $\mathcal{V}_t$, organized exploration history $\{H_k\}$, and the memory embedding $\mathcal{M}_t$. Here, $\{H_k\}$ records the visual tokens, reasoning traces, and actions of previous MLLM invocations, while $\mathcal{M}_t$ stores a compact memory of intermediate frames. When $s_t = 0$, LESA reuses the previous action and segmentation token to avoid redundant MLLM inference. The final mask $\hat{M}_t$ is decoded from $\hat{m}_t$ by a mask decoder.

## 4.2. State Controller

The State Controller $\mathcal{F}_{\text{SC}}$ is designed to support efficient online exploration. It maintains two complementary states: a compact hidden state $h_t$ that summarizes the current exploration progress, and a hierarchical memory $\mathcal{M}_t$ that preserves visual observations over different temporal spans. The controller updates these states online and predicts the trigger signal $s_t$ for selective MLLM invocation.

**Hierarchical Memory.** To preserve long-range visual information under a limited token budget, we maintain a multi-level memory bank $\mathcal{M}_t = \{\mathcal{M}_t^1, \ldots, \mathcal{M}_t^L\}$. New visual features are inserted into the lowest level $\mathcal{M}_t^1$. Once a level exceeds its capacity, the two oldest tokens are merged and promoted to the next level:

$$\mathcal{M}_{\text{merge}}^\ell = \text{Linear}\big(\text{SelfAttn}([\mathcal{M}_{\text{old}}^\ell; \mathcal{M}_{\text{older}}^\ell])\big). \quad (3)$$

This cascading update is applied recursively until all levels satisfy their capacity constraints. In this way, lower levels preserve fine-grained recent observations, while higher levels store compact summaries of distant history. The final memory embedding is formed by concatenating all levels with level-specific embeddings:

$$\mathcal{M}_t = [\mathcal{M}_t^L; \ldots; \mathcal{M}_t^1] + E_{\text{lvl}}. \quad (4)$$

**Exploration-Activated Calibration.** Whenever the MLLM produces a new exploration action $\hat{a}_t$, LESA calibrates the internal states to align them with the new exploration intent. The calibration function $f_{\text{ref}}$ first partially resets the hidden state toward the initial learnable queries:

$$\bar{h}_t = \alpha h_t + (1 - \alpha)h_{\text{init}}, \quad (5)$$

where $\alpha \in [0, 1]$ controls the reset degree. To suppress stale long-term context, it then retains only the lowest-level memory that contains recent observations:

$$\bar{\mathcal{M}}_t = \mathcal{M}_t^1. \quad (6)$$

Finally, the state and memory are refined by cross-attention conditioned on the expression and the newly planned action:

$$h_t, \mathcal{M}_t = \text{CrossAttn}\big([\bar{h}_t; \bar{\mathcal{M}}_t], [F_\mathcal{T}; F_{\hat{a}_t}]\big), \quad (7)$$

where $F_\mathcal{T}$ and $F_{\hat{a}_t}$ denote the text embeddings of the referring expression and the planned action, respectively. This

calibration injects goal- and action-aware semantics while reducing interference from outdated exploration history.

**Exploration-Aware State Update.** After calibration, the controller updates the hidden state as new frames arrive. At each timestep, the refined feature $\tilde{F}_t$ and memory embedding $\mathcal{M}_{t-1}$ are used to update $h_t$ through cross-attention:

$$h_t' = \text{CrossAttn}(h_{t-1}, [\mathcal{M}_{t-1}; \tilde{F}_t]). \quad (8)$$

To make the update sensitive to the current exploration goal, LESA predicts a semantic gate conditioned on the previous state, the expression, and the current action:

$$Z = \text{SelfAttn}([h_{t-1}; F_\mathcal{T}; F_{\hat{a}_{t-1}}]), \quad (9)$$

$$G_t = \sigma(W_g Z^{[h]} + b_g), \quad (10)$$

$$h_t = h_{t-1} + G_t \odot h_t', \quad (11)$$

where $Z^{[h]}$ denotes the output tokens corresponding to $h_{t-1}$. The gate controls how much new visual evidence should be integrated according to the ongoing exploration. Finally, a lightweight prediction head estimates the trigger probability:

$$s_t = \sigma(\text{MLP}(h_t)). \quad (12)$$

The trigger policy is trained with a BCE loss $\mathcal{L}_s$ using supervision that jointly reflects action transitions and object presence. Starting from the initial frame, LESA alternates between execution and reasoning phases. During execution, the controller continuously updates the hidden state and memory, *i.e.*, $(h_t, \mathcal{M}_t)$, from incoming frames and reuses the previous action and segmentation token. Once $s_t = 1$, the MLLM is invoked to produce a new action and `<SEG>` token, after which $f_{\text{ref}}$ immediately calibrates the controller state according to the new action and the referring expression. More implementation details are provided in the supplementary.

## 4.3. Role-Structured MLLM

We adopt a role-structured reasoning scheme to enable a single MLLM backbone to perform planning and segmentation with different functional behaviors. Instead of introducing separate models, we condition the same $\mathcal{F}_{\text{LLM}}$ on an explicit `<TASK>` token, which switches the functional role and regulates the reasoning pattern of the model.

Before exploration starts, the MLLM first analyzes the referring expression $\mathcal{T}$ and generates a global goal analysis. This global analysis is produced only once and reused throughout the whole exploration process, serving as a persistent high-level prior for subsequent planning and segmentation. At each triggered timestep, the input tokens are constructed by concatenating: **(i)** the expression $\mathcal{T}$ and the global goal analysis; **(ii)** organized history logs $\{H_k\}$ containing previous visual tokens, reasoning traces, and actions; **(iii)** hierarchical memory embeddings $\mathcal{M}_t$ summarizing recent and

*Table 2.* The human alignment experiments on 600 random collected pairs. * denotes advanced by designed prompts and in-context learning. All values are reported in percentage (%).

| Human Annotators Consistency | | | |
|---|---|---|---|
| Agreement | **95.2** | Kappa | **88.6** |
| **Judge Model** | | | |
| Method | Acc | F1 | Kappa $\kappa$ |
| LLaMA-3-8B-Instruct | 85.1 | 76.9 | 70.4 |
| Qwen3-4B-Instruct | 85.3 | 78.8 | 72.5 |
| Qwen2.5-7B-Instruct | 86.7 | 80.4 | 73.0 |
| **Qwen2.5-7B-Instruct*** | **89.6** | **82.2** | **75.1** |
| **Continuous Similarity Metric** | | | |
| Method | Acc | F1 | Spearman $\rho$ |
| BLEU | 66.6 | 53.3 | 56.5 |
| METEOR | 66.4 | 54.3 | 57.5 |
| BERTScore | 56.1 | 58.4 | 53.4 |
| **Qwen3-Embedding** | **70.5** | **68.2** | **67.4** |

long-term visual context; **(iv)** the current frame features; and **(v)** a role prompt with the <TASK> token.

When triggered by the State Controller, the shared $\mathcal{F}_{\text{LLM}}$ is invoked sequentially with two task roles. The Planner first reasons over the input token sequence to generate a structured reasoning trace and the next action $\hat{a}_t$. The Segmentor then conditions on the Planner output and produces the <SEG> token $\hat{m}_t$ for mask decoding. This design addresses the coupling between planning and segmentation by keeping both roles in a unified MLLM latent space, while using task-specific tokens to decouple their output behaviors: long-form reasoning for planning and stable short-form token generation for segmentation. The Planner is supervised by reference reasoning traces and actions via $\mathcal{L}_{llm}$, and the Segmentor is trained with $\mathcal{L}_{mask}$ and $\mathcal{L}_{dice}$ through the decoded mask.

The overall training objective combines role-structured language supervision, mask prediction, and trigger supervision:

$$\mathcal{L} = \mathcal{L}_{llm} + \lambda_{mask}\mathcal{L}_{mask} + \lambda_{dice}\mathcal{L}_{dice} + \lambda_s\mathcal{L}_s. \quad (13)$$

## 5. Experiments

### 5.1. Evaluation Setting

We acknowledge that evaluating fully active video exploration requires closed-loop execution with physical actuators. However, such settings incur high reproducibility costs and strong platform dependency, hindering public benchmarking. We therefore adopt the RVAS *val* set as a scalable proxy benchmark that decouples and evaluates planning and segmentation capacity under ground-truth observation

histories, assuming ideal actuator execution. At each inference timestep, the next observation $\mathcal{V}_{t+1}$ is provided from the recorded trajectory, rather than generated by executing $\hat{a}_t$. We argue that robust model behavior in this setting is essential for a successful active exploration agent.

For action evaluation, we compare the predicted action sequence with the annotated exploration policy. Since RVAS actions are open-set natural-language commands, exact matching or n-gram metrics such as BLEU and METEOR (Papineni et al., 2002; Banerjee & Lavie, 2005) cannot reliably capture semantic equivalence. We therefore use two complementary metrics. First, we compute frame-level semantic similarity using Qwen3-Embedding-8B (Zhang et al., 2025), and average it over each video as **Similarity**, which measures the overall alignment between predicted and human action policies. Second, we use Qwen2.5-7B-Instruct (Yang et al., 2025) as an evaluation judge to determine whether a predicted *new action* matches the ground-truth action within a $\pm 1s$ temporal window. We report **Acc-Any@1s**, which counts a success if at least one predicted new action matches the ground truth within the window, and **Acc-All@1s**, which requires all predicted new actions in the window to be correct. We use Acc-All@1s as the primary action accuracy metric because it penalizes unstable or oscillatory action predictions. We validate the reliability of the judge model and semantic similarity metric against human annotation in Table 2.

For segmentation evaluation, we follow prior RVOS works (Seonguk, 2020; Ding et al., 2023; 2025b; Ying et al., 2025b) and report $\mathcal{J}\&\mathcal{F}$. Specifically, $\mathcal{J}$ and $\mathcal{F}$ measure region IoU and contour accuracy respectively, with $\mathcal{J}\&\mathcal{F}$ as their average. However, RVAS poses a long-horizon foreground-localization challenge beyond conventional RVOS. The referred target is often absent before sufficient exploration, so video-level $\mathcal{J}\&\mathcal{F}$ can be dominated by target-absent frames and may overestimate models that frequently predict empty masks. Therefore, relying only on $\mathcal{J}\&\mathcal{F}$ may not fully reflect whether the target is correctly localized once it becomes visible.

To better evaluate whether the model successfully localizes the discovered target, we additionally report $\mathcal{J}_{fg}$, a foreground-only IoU computed on frames where the target is visible in the ground-truth annotations. This metric reduces the influence of trivial empty-mask predictions and focuses on foreground target localization after exploration. Together, $\mathcal{J}\&\mathcal{F}$ evaluates overall video-level segmentation quality, while $\mathcal{J}_{fg}$ more directly reflects the model's ability to segment the referred object once it becomes observable.

### 5.2. Main Results

As shown in Table 3, we benchmark representative referring video object segmentation (RVOS) and related video un-

*Table 3.* The main experiments results on RVAS dataset of two primary evaluation dimensions. AccY and AccA refers to the metrics Acc-Any@1s and Acc-All@1s, respectively. **Bold** and underline indicate the largest and the second largest values.

| Method | Type | Language Model | FPS | Planning | | | Segmentation | | | |
|---|---|---|---|---|---|---|---|---|---|---|
| | | | | Sim | AccY | AccA | $\mathcal{J}$ | $\mathcal{F}$ | $\mathcal{J}\&\mathcal{F}$ | $\mathcal{J}_{fg}$ |
| LoSh | RVOS | ✗ | ✗ | ✗ | ✗ | ✗ | 38.6 | 40.1 | 39.3 | 7.0 |
| ReferDINO | RVOS | ✗ | ✗ | ✗ | ✗ | ✗ | 52.3 | 53.9 | 53.1 | 12.7 |
| VideoLLM-Online | Streaming | Llama2-7B-Chat | 1.65 | 47.9 | 15.4 | 13.1 | ✗ | ✗ | ✗ | ✗ |
| Dispider | Streaming | Qwen2-7B | 1.03 | 46.4 | 23.9 | 7.3 | ✗ | ✗ | ✗ | ✗ |
| QwenVL3-8B | MLLM | QwenVL3-8B-Instruct | 0.19 | 49.4 | 40.4 | 6.9 | ✗ | ✗ | ✗ | ✗ |
| InternVL3-8B | MLLM | InternVL3-8B | 0.21 | 50.7 | **42.3** | 7.2 | ✗ | ✗ | ✗ | ✗ |
| VISA | LLM-RVOS | Chat-UniVi-7B | 0.22 | 43.8 | 28.6 | 5.2 | 42.1 | 43.8 | 43.0 | 6.6 |
| InstructSeg | LLM-RVOS | Mipha-3B | 0.18 | 40.2 | 20.9 | 4.4 | 28.3 | 29.9 | 29.1 | 10.2 |
| GLUS-7B | LLM-RVOS | LISA-7B-v1 | 0.24 | 45.9 | 25.9 | 8.1 | 54.9 | 55.7 | 55.3 | 7.1 |
| Sa2VA-8B | Unified | InternVL2.5-8B | 0.16 | 48.5 | 31.9 | 14.9 | 55.4 | 56.4 | 55.9 | 8.6 |
| Sa2VA-8B* | Unified | InternVL2.5-8B | 0.14 | 48.4 | 31.6 | 15.0 | 45.5 | 47.1 | 46.3 | 15.7 |
| UniPixel-7B | Unified | Qwen2.5-VL-7B-Instruct | 0.11 | 46.5 | 27.8 | 11.4 | 52.9 | 53.8 | 53.4 | 9.0 |
| UniPixel-7B* | Unified | Qwen2.5-VL-7B-Instruct | 0.11 | 46.5 | 28.0 | 11.4 | 44.0 | 45.4 | 44.7 | 16.2 |
| **LESA (Ours)** | RVAS | InternVL2.5-8B | **2.21** | **52.1** | 25.0 | **22.2** | **58.2** | **60.6** | **59.4** | **22.7** |

derstanding methods under a capability-aligned evaluation protocol. For conventional RVOS methods without language models (Yuan et al., 2024; Liang et al., 2025), we evaluate only offline segmentation performance. Streaming video understanding models (Chen et al., 2024a; Qian et al., 2025) are tested following their original streaming paradigms. For multimodal large language models (Zhu et al., 2025), at each time step $t$, we prompt the model in a QA format to decide the next action based on past observations, enabling streaming responses. RVOS methods that perform LLM-based segmentation on sparse frames (Yan et al., 2024; Wei et al., 2025; Lin et al., 2025) are evaluated by isolating their language models and applying their default protocol, i.e., segmenting once at the end of the video. Unified methods (Yuan et al., 2025; Liu et al., 2025) are additionally evaluated under a streaming planning-and-segmentation setting (denoted as *), where the model decides online whether to segment, closer to active exploration. Our LESA follows the same online protocol. Except for Dispider, whose training script is unavailable, all methods are trained on the RVAS dataset for three epochs for fair comparison.

As shown in Table 3, prior segmentation approaches exhibit clear limitations in long-horizon exploration. Due to sparse sampling and mask propagation, these methods achieve high $\mathcal{J}\&\mathcal{F}$ under the conventional segment-once protocol but suffer from low $\mathcal{J}_{fg}$ (often below 10%), indicating severe target omission in complex scenarios. Under the streaming segmentation protocol*, unified models attend to and process all frames to decide when to segment, resulting in improved $\mathcal{J}_{fg}$ (15.7% on Sa2VA and 16.2% on UniPixel). However, this gain comes at the cost of substantial $\mathcal{J}\&\mathcal{F}$

*Table 4.* The oracle study in which the exact decision time steps are provided to evaluate the model's single-turn planning capacity.

| Model | AccY | AccA | Acc-oracle |
|---|---|---|---|
| Sa2VA-8B* | 31.6 | 15.0 | 20.8 (+5.8%) |
| QwenVL3-8B | 40.4 | 6.9 | 19.9 (+13.0%) |
| InternVL3-8B | 42.3 | 7.2 | 21.9 (+14.7%) |
| **LESA (Ours)** | 25.0 | 22.2 | 24.6 (+2.4%) |

degradation, caused by premature segmentation before sufficient exploration. In contrast, LESA jointly models planning trajectories and short-term memory for segmentation, enabling reliable object discovery and fine-grained visual perception, resulting in significantly improved segmentation performance of 59.4% $\mathcal{J}\&\mathcal{F}$ and 22.7% $\mathcal{J}_{fg}$. For exploration planning, LESA produces more stable predictions under dynamically changing environments, allowing for effective determination of when to switch exploration policies and generate consistent signals. The perception-reasoning decoupling framework achieves higher Sim (52.1%) and Acc-All (22.2%), remaining competitive with prior MLLMs while achieving an approximately $10\times$ inference speedup.

By comparison, prior streaming video methods, which mainly focus on observation summarization and QA, show limited capability in goal-oriented exploration (15.4% Acc-Any and 13.1% Acc-All) and lack support for fine-grained segmentation. Furthermore, oracle experiments in Table 4 demonstrate that even with precise decision points and complete observation histories, existing models still underperform LESA (21.9% vs. 24.6%), highlighting their difficulty in effectively collecting and integrating long- and short-term exploration-aware information.

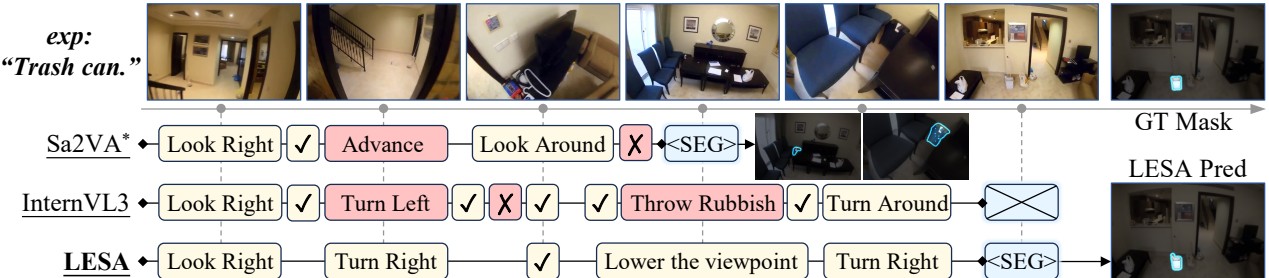

*Figure 4.* Qualitative results of Sa2VA*, InternVL3, and our proposed LESA on the RVAS sample. Sa2VA* mistakenly segments a similar object in advance of full exploration, while InternVL3 lacks exploration understanding and tends to visit a previously explored direction.

*Table 5.* Ablation study on history strategy. Response means dynamically organizing the history utilizing exploration process.

| ID | History | Recent | AccA | $\mathcal{J}\&\mathcal{F}$ | $\mathcal{J}_{fg}$ |
|---|---|---|---|---|---|
| I | Uniform | ✗ | 17.1 | 54.8 | 16.3 |
| II | Uniform | Frame | 19.4 | 56.2 | 20.6 |
| III | Response | Frame | 20.5 | 57.0 | 20.9 |
| IV | Response | Memory | **21.3** | **57.4** | **21.6** |

*Table 6.* Ablation study on State Controller. *G.* represents gating. Hybrid means utilizing Frame + Memory as state update source.

| ID | Hidden State | Source | AccA | $\mathcal{J}\&\mathcal{F}$ | $\mathcal{J}_{fg}$ |
|---|---|---|---|---|---|
| I | Memory | Frame | 14.9 | 46.2 | 12.9 |
| II | Query | Frame | 18.9 | 53.2 | 19.7 |
| III | Query+G. | Frame | 19.2 | 54.1 | 20.4 |
| IV | Query+G. | Hybrid | **21.6** | **57.2** | **22.3** |

Figure 4 presents a representative example comparing our LESA with the baseline methods Sa2VA* and InternVL3. We expose the core deficiencies of prior approaches with a simple scene and expression: First, they exhibit inadequate modeling of the exploration process. InternVL3 tends to revisit previously explored directions (Turn Left), whereas Sa2VA* prematurely commits to a visually ambiguous object under insufficient exploration, resulting in incorrect segmentation. Besides, dense prediction induces control inconsistency and substantially amplifies susceptibility to hallucinations (Throw rubbish). In contrast, LESA demonstrates stable and reliable exploration planning and robust segmentation grounded in a thorough exploration process.

### 5.3. Ablation Studies

**History Processing Strategy.** As shown in Table 5, organizing long-term history by planning decisions and encoding short-term observations into memory embeddings consistently outperforms uniform sampling and direct use of recent frames. Compared with uniformly sampling previous frames (I, II), decision-aware organization (III) better leverages the MLLM to capture critical cues and retain informative reasoning traces, improving Acc-All from 17.1% and 19.4% to 20.5%. Hierarchical memory encoding further extends the temporal receptive field under the same token budget, lifting Acc-All to 21.3% and $\mathcal{J}_{fg}$ to 21.6% (IV).

**State Controller Design.** We analyze the design choices of the state controller in Table 6. Predicting the event score directly from memory embeddings without a query token (I) performs poorly. Introducing a hidden state $h_t$ to summarize high-level semantics (II) and together with an exploration-

aware gate attention (III) leads to a consistent Acc-All (14.9 % to 19.2 %) and $\mathcal{J}_{fg}$ gains (12.9% to 20.4%). Jointly updating the hidden state with memory embeddings and new observations (IV) further enhances contextual awareness, achieving 21.6% Acc-All and 22.3% $\mathcal{J}_{fg}$.

## 6. Conclusion

We introduce Referring Video Active Exploration and Segmentation (RVAS), a new task that focuses on active goal-driven exploration beyond passive perception of conventional RVOS. To support this task, we build a large-scale egocentric dataset with diverse query types and fine-grained reasoning annotations. We further propose LESA, an efficient framework that integrates a lightweight state controller with a role-structured MLLM to reduce streaming latency and maintain history and memory in an exploration-aware manner. Experiments show that LESA achieves superior segmentation and planning performance over strong baselines, while improving inference efficiency significantly.

**Limitations and Future Work.** The current RVAS dataset is designed as a reproducible testbed for active exploration prior to fully embodied deployment. Although the evaluation relies on pre-recorded egocentric trajectories with idealized action execution, this design inevitably leaves a gap from real-world closed-loop settings. Future work could extend RVAS toward closed-loop evaluation by introducing a standardized language-driven executor, such as a promptable VLA-based policy, within a simulation environment that exposes a fixed high-level plan-input interface. Such a design would enable closed-loop control while preserving the reliability and reproducibility of the evaluation protocol.

## Acknowledgment

This work was supported by the National Natural Science Foundation of China (NSFC) under Grant No. 62472104 and the Science and Technology Commission of Shanghai Municipality under Grant No. 25511103600.

## Impact Statement

This work takes a step toward connecting high-level language understanding with fine-grained visual grounding in sequential, egocentric video settings. By formalizing the RVAS task, we study how implicit user intents (e.g., "Check the time") can be addressed through active visual search over video streams, without assuming explicit spatial supervision or static inputs. While RVAS is evaluated on pre-recorded videos, the task formulation and proposed LESA framework may serve as a useful visual perception and reasoning component for future embodied systems.

LESA further provides a practical architectural reference for integrating large multimodal models into latency-sensitive streaming scenarios by selectively invoking high-level reasoning. We acknowledge that the current evaluation does not capture the full uncertainty of real-world interaction. Future work should explore online and interactive settings, address the sim-to-real gap, and carefully consider privacy concerns associated with egocentric vision before deployment in real-world environments.

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

## A. RVAS Dataset

### A.1. Detailed statistics

We provide more detailed statistics of the proposed RVAS dataset. RVAS contains videos spanning five different types of scenes: desktop, single-room indoor, multi-room indoor, outdoor, and robotics. The distribution of the scenes is shown in Figure 5.

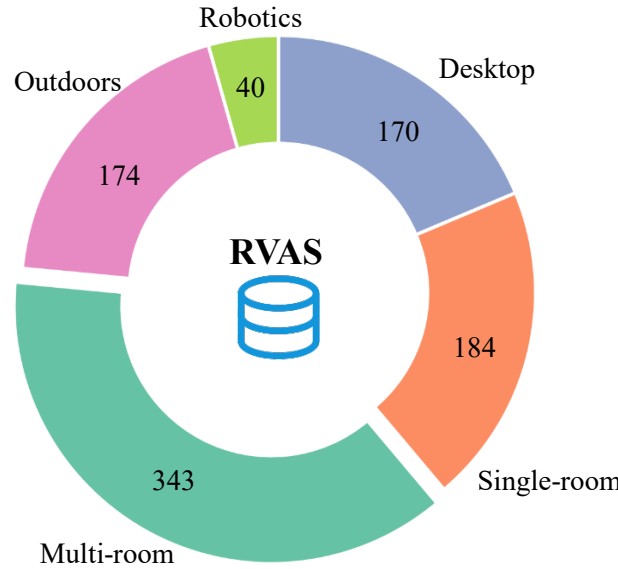

*Figure 5.* The distribution of the scene types in RVAS.

Considering the application scenarios of active exploration and the inherent bias of publicly available video data, multi-room scenes constitute the majority of the RVAS dataset (37.7%). In contrast, outdoor scenes, which are typically highly complex, and desktop or single-room scenes with relatively limited spatial extent, are more evenly distributed, each accounting for approximately 20% of the dataset. In addition, a small portion of first-person videos captured from real robot manipulation scenarios is included to further enhance the diversity and the practical utility of our dataset.

We further show the target-object category distribution in Figure 6. The balanced and diverse composition across multiple semantic superclasses makes RVAS better aligned with the real-world applications. Objects related to built environments and infrastructure, electronic devices, and household-related categories (including furniture, appliances, and daily necessities) constitute a substantial portion of the dataset, reflecting the predominantly indoor and human-centric nature of the collected scenes.

At the same time, categories such as food, drink, cleaning and hygiene items, and kitchenware are consistently represented, ensuring coverage of fine-grained, frequently manipulated objects commonly encountered in everyday

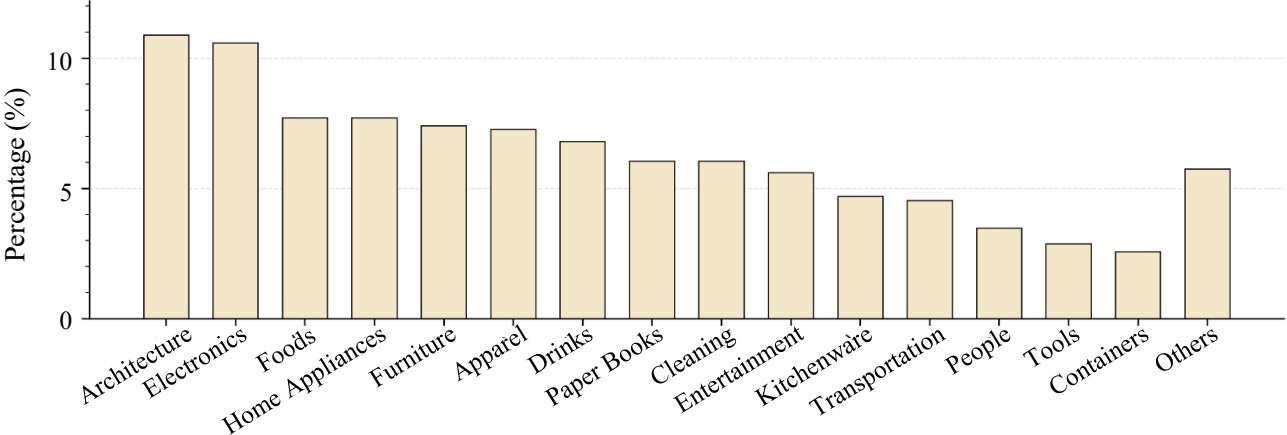

*Figure 6.* The distribution of target object categories in the RVAS dataset. The dataset focuses on object categories commonly encountered in daily life, enabling better generalization to household environments and everyday scenarios.

activities. Overall, this distribution highlights the dataset's ability to support active perception and exploration tasks across a wide range of object types, while avoiding over-concentration on a single category.

We have reported three different types of language expressions involved in RVAS, containing 1,978 object expressions, 1,458 intent expressions, and 1,197 instruction expressions. These new expression types pose a new challenge to both implicit reasoning and language-guided efficient exploration. Several representative examples are provided below to illustrate the diversity and characteristics of these expressions.

To further illustrate the diversity of environments and exploration behaviors in RVAS, Figure 7 presents representative samples spanning all five scene categories. These examples cover a wide range of agent behaviors, from desktop manipulation and single-room search to multi-room navigation, outdoor traversal, and robotic interaction, demonstrating that RVAS jointly stresses commonsense reasoning, long-horizon planning, and fine-grained object localization across heterogeneous scenes.

### A.2. Annotation Pipelines

In this section, we provide a detailed description of the data annotation pipeline which is summarized in the main paper. We organized a team of well-trained annotators and adopted a three-stage pipeline to attain high-quality annotations:

In the first stage, we asked annotators to generate three different types of language expressions, spanning object, intent, and instruction, for the selected targets. Another group of annotators was responsible for re-tracking the video and re-moving unreasonable and ambiguous expressions, ensuring the correct correspondence between objects and expressions.

In the second stage, annotators were required to identify the decision point and annotate the corresponding action decisions based on given expressions, using a streaming annotation tool without access to future frames. Annotators were also required to provide a brief explanation describing the rationale behind each decision. Each candidate's video was assigned to at least three annotators. Only those videos with a majority agreement on both the decision points and the action decisions are retained to keep the most annotated action sequences natural and reasonable. In other words, videos containing clearly unreasonable exploration actions or trajectories (*e.g.* for the expression "I want to cut some vegetables.", walking into the bathroom can be unreasonable) are filtered out at this stage.

In the final stage, full mask sequences and reference reasoning traces are annotated using an offline interactive annotation platform. Object masks are annotated with the assistance of the SAM2 model. Annotators generate reasoning traces by jointly analyzing the given expression, historical frames, and brief decision explanations from the previous stage, following a structured `goal analysis`, `observation`, `inference`, `planning` template. All reasoning traces are subject to quality control via training with curated exemplars and subsequent audits. We show an annotated reasoning trace randomly selected from RVAS dataset in Figure 8.

In particular, to enable more accurate evaluation of exploration planning and to improve robustness to semantically similar yet differently expressed actions, we preserve and manually augment annotations with equally reasonable alternative actions at each decision point. As illustrated in Figure 9, these alternative annotations fall into two complementary categories: **(i)** *semantically equivalent rephrasings* of the same action at different granularities (*e.g.*, *Move forward*, *Go around the table*, and *Go to the shelf directly*

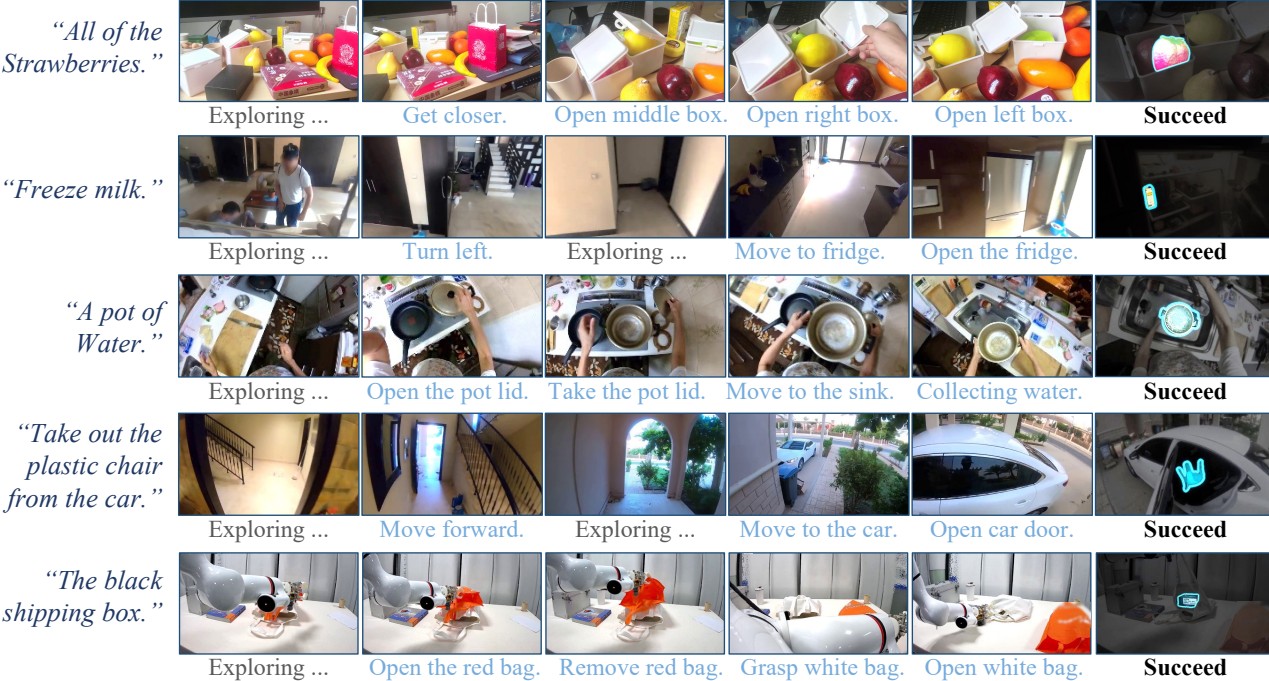

*Figure 7.* More representative examples from RVAS across the five scene categories (*i.e.*, desktop, single-room indoor, multi-room indoor, outdoor, and robotics). Each row corresponds to a different scene type. Given a goal expression on the left, the agent produces a sequence of actions under streaming observations, where *"Exploring ..."* indicates intermediate exploration phases with some low-level steps omitted for brevity.

*ahead* all describe the same underlying behavior), which mitigate the sensitivity of LLM-based judges to linguistic surface variations; and **(ii)** *alternative exploration strategies* corresponding to different yet equally valid trajectories (*e.g.*, approaching either the yellow packaged beverage or the paper stapler when both could lead to the target), applied only when no clear visual cue dictates a single path.

This design partially alleviates the limitation of strictly treating pre-recorded trajectories as ground-truth labels during evaluation. By allowing models to predict alternative actions that are consistent with the overall exploration intent, rather than enforcing exact matching to a single recorded action, the evaluation more faithfully reflects the quality of model behavior and the multi-path nature of real-world exploration.

## B. LESA Framework

### B.1. Additional Method Details

We supplement the main paper with additional details on the update mechanism of the hierarchical memory, the formulation of the exploration-activated calibration function $f_{\text{ref}}$, and the training targets for the binary trigger score.

**Implementation Details of Hierarchical Memory.** In the hierarchical memory bank $\mathcal{M}_t = \{\mathcal{M}_t^1, \ldots, \mathcal{M}_t^L\}$, each level $\mathcal{M}_t^\ell$ has a predefined capacity $C^\ell$ fixed during both training and inference. The highest level $\mathcal{M}_t^L$ serves as an archive level with $C^L = 1$, storing the most compressed representation of distant history. The merge-and-promote update formalized in the main paper is applied recursively until all levels satisfy their capacity constraints; for levels that are not fully populated, zero vectors $\mathbf{0}$ are used for padding before constructing the final memory embedding.

**Implementation Details of Exploration-Activated Calibration.** As introduced in the main paper, the calibration module $f_{\text{ref}}$ follows a reset-and-refine strategy whenever the MLLM produces a new exploration action: the hidden state is partially reset toward the initial learnable queries $h_{\text{init}}$ with coefficient $\alpha \in [0, 1]$, stale long-term memory is discarded by retaining only the lowest-level $\mathcal{M}_t^1$, and the reset state and retained memory are then refined through cross-attention with the expression and action embeddings $[F_{\mathcal{T}}; F_{\hat{a}_t}]$. This operation suppresses outdated context from previous exploration clips while preserving recent observations and injecting goal- and action-aware semantics into the controller state.

**Trigger Supervision.** The trigger prediction head is trained with binary supervision over sampled frame sequences, where positive labels mark frames at which a new MLLM reasoning step is required, determined jointly by action tran-

*Table 7.* Representative examples of language expressions in RVAS.

| Expression | Type | Characteristic |
|---|---|---|
| Kettle. | Object | **Minimal object reference** without attributes or context, requiring basic object recognition. |
| The keys hang on the door. | Object | Object description with an **explicit spatial relation**, involving spatial grounding between entities. |
| The highest man in the room. | Object | **Relational description** that requires full exploration over multiple candidates. |
| I'm going to call my mom. | Intent | **High-level intent** without specifying the required object or action sequence. |
| What is the time right now? | Intent | A **more implicit intent** requiring an in-depth reasoning. |
| I just spilled some water on the floor. | Intent | **Ambiguous intent** which allows multiple valid object choices *e.g.* a mop or a cloth. This makes the exploration planning more complex. |
| Please go upstairs and turn off the TV. | Instruction | **Direct instruction** that guides the model to explore. |
| Search the current room for my headphones. | Instruction | Instruction with an **explicit scope constraint**, defining a bounded exploration region, thus enabling the empty-target samples in the dataset. |

**Reasoning Trace Example.**
```
The task involves locating the small
TV, typically positioned on the cabinet
in the living room, though it might
occasionally be in the bedroom.  Given
the higher likelihood of its presence in
the living room, the initial step should
be to identify and proceed to the living
room to conduct the search.
The reflection in front of me shows
a mirror, and the walls are lined
with dark tiles, creating a sleek and
modern appearance.  This setup strongly
indicates that I'm in a bathroom.  The
tiles are uniformly placed, and the
lighting above casts a bright glow,
enhancing the room's clean and organized
look.  Given that bathrooms typically
don't have televisions, it seems unlikely
that I will find one here.  To proceed
effectively, I should take a moment
to thoroughly observe my surroundings,
noting the layout and any exits.  This
will help me decide the best direction
to move in next as I continue my search
elsewhere.
```

*Figure 8.* An example annotated reasoning trace.

where $s_t$ is the predicted trigger probability at frame $t$.

**Hyperparameters.** Unless otherwise specified, the memory capacities are set to $[6, 4, 4, 1]$ across four levels, and each memory token and the video state $h_t$ have dimensionality 128. The trigger score threshold during inference is fixed to 0.5.

### B.2. Implementation Details

We adopt InternVL2.5-8B (Chen et al., 2024b) as the MLLM backbone and build a multi-stage training pipeline upon the xtuner framework. Following prior unified models (Yuan et al., 2025), we employ SAM2 (Ravi et al., 2024) as the segmentation head. In the first stage, we disable the State Controller, train only a small subset of SAM2 parameters, and apply LoRA-based fine-tuning (Hu et al., 2022) to the backbone. This stage is trained on large-scale referring image/video segmentation datasets (Kazemzadeh et al., 2014; Yu et al., 2016; Seonguk, 2020; Ding et al., 2023; Yan et al., 2024) to initially align features between the MLLM and the segmentation head. In the second stage, except for partial segmentation head parameters, we optimize all State Controller parameters and adopt a higher-rank LoRA fine-tune of the MLLM on the RVAS dataset, jointly training all the components end-to-end. For preprocessing, inputs are resized to $(448, 448)$ for the MLLM processor and $(1024, 1024)$ for the SAM2 image encoder, following their default protocol. For LoRA parameters, we set $r$ to 128 and the LoRA scaling factor to 256 in both stages. We

sitions and target-object presence changes. The per-frame supervision signal $y_t$ is composed of two complementary components.

**Temporal Proximity Score ($P_t$):** To anticipate action transitions, we apply a Gaussian-like decay centered at each ground-truth transition timestamp $\hat{t}$, where $P_t = \exp\left(-\frac{(t-\hat{t})^2}{2\sigma^2}\right)$. This provides a smooth optimization gradient around decision boundaries.

**Visual Saliency Prior ($V_t$):** To prioritize frames with high informational value, we scale the relative area of the target object's mask to the range $[0.5, 1]$. Let $A_t$ be the mask area at frame $t$; then $V_t = 0.5 \cdot (1 + A_t / \max_t A_t)$. This encourages the controller to trigger MLLM reasoning when the object is prominently visible, facilitating more reliable segmentation.

The final supervision is set as $y_t = \max(P_t, V_t)$, and the trigger head is optimized via a binary cross-entropy loss:

$$\mathcal{L}_s = -\frac{1}{T} \sum_{t=1}^{T} [y_t \log s_t + (1 - y_t) \log(1 - s_t)], \quad (14)$$

**Exp:** *"The front door."*      **Exp:** *"Refrigerator."*      **Exp:** *"Bake the bread."*

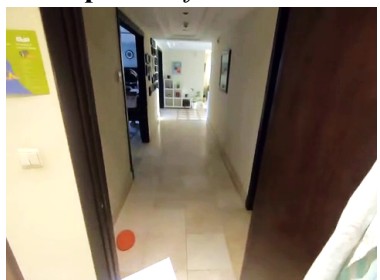 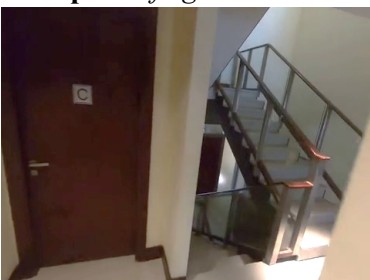 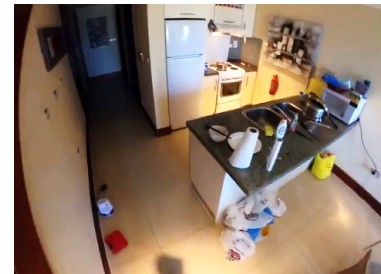



Keep Moving Forward.      Turn right.      Enter the kitchen.

Navigate to the living room ahead.      Go downstairs via stairs.      Go around the desktop.

Check the room on the left.      Check the front room.      Turn right.



*Figure 9.* Alternative action annotations at a single decision point. **Green**: the executed ground-truth action. **Blue**: additional valid actions *treated as correct* during evaluation, including semantically equivalent rephrasings and alternative exploration strategies.

employ AdamW optimizer (Loshchilov & Hutter, 2017) with a learning rate of $4 \times 10^{-5}$. Loss weights are set to $\lambda_{mask} = 1.0$, $\lambda_{dice} = 2.0$, and $\lambda_s = 0.5$. Training runs for three epochs with a per-device batch size of 1 and 8 gradient accumulation steps. All experiments are conducted on 8 NVIDIA A100 40GB GPUs.

## C. RVOS Experiments

As shown in Table 8, we further evaluate LESA on prior RVOS benchmarks, including MeViS (Ding et al., 2023), ReasonVOS (Bai et al., 2024), and ReVOS (Yan et al., 2024). We observe that LESA does not outperform the current strongest RVOS method, UniPixel (Liu et al., 2025), which is trained on substantially larger-scale image and video datasets with a stronger backbone, and is specifically optimized for conventional RVOS settings. This performance gap is expected, as most core components of LESA—including the state controller, exploration history organization, and role-structured MLLM reasoning—are designed for active exploration and decision-aware segmentation, and cannot be fully exercised or effectively trained under standard RVOS protocols. Moreover, the available training data do not support zero-shot generalization of these exploration-centric modules to other datasets.

Despite this mismatch in task formulation, LESA consistently performs on par with or slightly better than Sa2VA (Yuan et al., 2025), a unified model built upon the same backbone, and GLUS (Lin et al., 2025), a representative recent RVOS method, ranking second only to UniPixel across benchmarks. This result indicates that LESA does not sacrifice fundamental segmentation capability, even though its design is not tailored for traditional RVOS. Instead, the performance reflects a deliberate trade-off: LESA prioritizes exploration-aware reasoning and long-horizon decision modeling over optimizing static RVOS benchmarks.

These results suggest that LESA maintains competitive segmentation quality while extending beyond the scope of conventional RVOS toward more general and realistic active exploration scenarios.

## D. Reliability of the LLM Judge for Action Evaluation

Since RVAS actions are open-set natural-language commands, traditional exact-match or n-gram metrics such as BLEU and METEOR cannot reliably capture semantic equivalence. We therefore adopt an instruction-aware embedding similarity metric and a structured LLM judge as our primary evaluation tools for predicted action sequences. In this section, we provide additional analyses on the reliability and potential bias of the LLM judge.

**Setup.** The judge is calibrated with carefully designed prompts and in-context examples that explicitly cover challenging linguistic patterns including antonyms, directional conflicts, negations, and granularity mismatches. To ensure reproducibility, we adopt an open-source backbone (Qwen2.5-7B-Instruct (Yang et al., 2025)), use deterministic decoding with temperature = 0, and parse structured outputs to obtain binary match decisions. Reliability against human annotation is reported in Table 2, where the judge achieves an accuracy of 89.6% and a Cohen's $\kappa$ of 75.1% on 600 randomly sampled human-labeled pairs.

**Comparison Against Traditional Metrics.** Table 9 provides a fine-grained comparison between the LLM judge and traditional metrics on challenging action pairs. Traditional metrics fail in two complementary directions: on the rejection side (Group A), they assign misleadingly high scores to semantically opposite pairs (BERT 0.86, METEOR 0.69) and miss verb antonymy, directional conflicts, and negations; on the acceptance side (Group B), they penalize valid

*Table 8.* Comparison of LESA against representative RVOS methods on MeViS, ReasonVOS, and ReVOS. All values are reported in percentage (%).

| Model | Backbone | MeViS (valid_u) | | | ReasonVOS | | | ReVOS | | |
|---|---|---|---|---|---|---|---|---|---|---|
| | | $\mathcal{J}$ | $\mathcal{F}$ | $\mathcal{J}\&\mathcal{F}$ | $\mathcal{J}$ | $\mathcal{F}$ | $\mathcal{J}\&\mathcal{F}$ | $\mathcal{J}$ | $\mathcal{F}$ | $\mathcal{J}\&\mathcal{F}$ |
| GLUS-7B | LISA-7B-v1 | 56.2 | 63.5 | 59.9 | 47.5 | 52.4 | 49.9 | 52.4 | 57.3 | 54.9 |
| Sa2VA-8B | InternVL2.5-8B | 54.8 | 63.5 | 59.1 | 51.9 | 58.5 | 55.2 | – | – | 57.6 |
| UniPixel-7B | Qwen2.5-VL-7B-Instruct | **58.3** | **65.0** | **61.7** | **56.5** | **63.0** | **59.7** | **61.7** | **65.7** | **63.7** |
| **LESA (Ours)** | InternVL2.5-8B | 56.6 | 63.7 | 60.1 | 52.9 | 59.1 | 56.0 | 54.8 | 60.5 | 57.7 |

equivalences expressed with different surface forms, with METEOR dropping to 0.30 on average and to 0.00 for "scan the area" vs. "look around". In contrast, the LLM judge correctly resolves all 16 challenging semantic distinctions, matching human labels in every case.

**Reproducibility and Bias Analysis.** Table 10 verifies reproducibility and analyzes the conservative tendency of the judge. Under deterministic decoding, all 18 test cases produce identical outputs across 10 repeated runs, indicating that the LLM judge is fully reproducible in our evaluation setup. The bottom rows highlight two borderline cases where the judge rejects pairs that a human annotator may consider equivalent: "remove" vs. "move away" (verb-level mismatch) and "check behind the door" vs. "open the door and look" (implicit vs. explicit decomposition). This conservative tendency increases evaluation strictness and is partially mitigated by the alternative valid actions mechanism described in Section A, which provides multiple ground-truth references at each decision point to better tolerate valid action variations.

*Table 9.* LLM judge vs. traditional metrics on challenging action pairs. ✔ = match, ✗ = mismatch. In Group A, traditional metrics assign misleadingly high scores (BERT 0.86, METEOR 0.69) to semantically *opposite* pairs, failing to capture verb antonymy, directional conflict, and negation. In Group B, they penalize valid equivalences expressed with different wording (METEOR drops to 0.00 for "scan the area" vs. "look around"). The LLM judge achieves **16/16 agreement with human labels**, correctly resolving all challenging semantic distinctions in both groups.

| Type | GT Action | Pred Action | Human | LLM | BERT | METEOR |
|---|---|---|---|---|---|---|
| *Group A: Surface-similar but semantically different (should be rejected)* | | | | | | |
| Antonym | open the door | close the door | ✗ | ✗ | 0.78 | 0.63 |
| Antonym | turn on the light | turn off the light | ✗ | ✗ | 0.88 | 0.64 |
| Direction | turn left | turn right | ✗ | ✗ | 0.95 | 0.25 |
| Direction | move the box to the left | move the box to the right | ✗ | ✗ | 0.98 | 0.83 |
| Object | pick up the red box | pick up the blue box | ✗ | ✗ | 0.98 | 0.75 |
| Object | grab the bottle | grab the tissues | ✗ | ✗ | 0.84 | 0.63 |
| Negation | open the drawer | do not open the drawer | ✗ | ✗ | 0.79 | 0.92 |
| Negation | move forward | stop moving forward | ✗ | ✗ | 0.68 | 0.89 |
| | | **Group A Avg.** | | **8/8** | **0.86** | **0.69** |
| *Group B: Surface-different but semantically equivalent (should be accepted)* | | | | | | |
| Synonym | grab the cup | pick up the cup | ✔ | ✔ | 0.87 | 0.61 |
| Synonym | move forward | walk ahead | ✔ | ✔ | 0.73 | 0.25 |
| Paraphrase | look around the room | observe the surroundings | ✔ | ✔ | 0.78 | 0.13 |
| Paraphrase | retract the arm | pull the arm back | ✔ | ✔ | 0.74 | 0.61 |
| Granularity | turn left to check the room | turn to the left | ✔ | ✔ | 0.75 | 0.35 |
| Granularity | navigate to table & grab cup | grab the cup from the table | ✔ | ✔ | 0.81 | 0.32 |
| Rephrasing | get rid of the obstacle | move the box away | ✔ | ✔ | 0.62 | 0.10 |
| Rephrasing | scan the area | look around | ✔ | ✔ | 0.66 | 0.00 |
| | | **Group B Avg.** | | **8/8** | **0.74** | **0.30** |

*Table 10.* Reproducibility and bias analysis. Under deterministic decoding (temperature = 0, greedy search), all 18 test cases produce **identical outputs across 10 repeated runs** (0 unstable cases). The bottom rows show two borderline cases where the judge conservatively rejects pairs that a human annotator may consider equivalent: "remove" vs. "move away" (verb-level mismatch) and "check behind the door" vs. "open the door and look" (implicit vs. explicit decomposition). This conservative tendency increases evaluation strictness and is mitigated in practice by our *alternative valid actions* mechanism (Section A), which provides multiple GT references at each decision point.

| Type | GT Action | Pred Action | Human | LLM | Consistency |
|---|---|---|---|---|---|
| *Representative cases from reproducibility test (18 total)* | | | | | |
| Antonym | open the door | close the door | ✗ | ✗ | 10/10 |
| Direction | turn left | turn right | ✗ | ✗ | 10/10 |
| Object | pick up the red box | pick up the blue box | ✗ | ✗ | 10/10 |
| Negation | open the drawer | do not open the drawer | ✗ | ✗ | 10/10 |
| Synonym | grab the cup | pick up the cup | ✔ | ✔ | 10/10 |
| Paraphrase | look around the room | observe the surroundings | ✔ | ✔ | 10/10 |
| Rephrasing | get rid of the obstacle | move the box away | ✔ | ✔ | 10/10 |
| Preposition | go around the table | go to the table | ✗ | ✗ | 10/10 |
| *Borderline cases illustrating conservative bias* | | | | | |
| Verb overlap | remove the box from table | move the box away | ✔ | ✗ | 10/10 |
| Impl. action | check behind the door | open the door and look | ✔ | ✗ | 10/10 |

