# OpenReview forum: "RVAS: Referring Video Active Exploration and Segmentation"
_ICML.cc/2026/Conference — ICML 2026 regular_

### Official Review · Reviewer_XzAR · 2026-03-05

**Soundness:** 3
**Presentation:** 3
**Significance:** 2
**Originality:** 2
**Overall Recommendation:** 4
**Confidence:** 4

**Summary:**

The paper introduces a new task called Referring Video Active Exploration and Segmentation (RVAS). Unlike traditional Referring Video Object Segmentation (RVOS) which assumes passive perception of the entire video, RVAS requires the model to perform online planning (predicting exploration actions) and segmentation simultaneously based on streaming visual inputs and open-ended language queries. To support this, the authors contribute a dataset containing 911 videos with annotated actions and reasoning traces. They propose LESA, a framework that utilizes a lightweight State Controller to efficiently manage streaming inputs and trigger a heavier MLLM (InternVL) for planning and segmentation only when necessary.

**Compliance With Llm Reviewing Policy:**

Affirmed.

**Final Justification:**

See in the Rebuttal Acknowledgement section

**Key Questions For Authors:**

1. The framework relies heavily on SAM2 for mask generation. Given the rapid evolution of segmentation models, have you considered how LESA would integrate with next-generation models, such as the potential SAM3?


2. To what extent does the "Action Planning" actually help "Segmentation" in your inference setup? Since the video feed is fixed regardless of the predicted action, if you remove the action prediction head (turning LESA into a passive streaming segmentation model with a state controller), how much does the segmentation metric actually drop?

**Limitations:**

The authors discuss limitations regarding the use of pre-recorded videos in the impact statement.

**Strengths And Weaknesses:**

Strengths:

1. The construction of the RVAS dataset is a solid contribution. The inclusion of "intent" and "instruction" based queries, along with manually annotated reasoning traces and action steps, adds value compared to traditional object-centric RVOS datasets. This data could be useful for training agents in video predictive tasks.

2. The design of LESA, specifically the decoupling of the lightweight State Controller from the heavy MLLM backbone, is a logical approach for streaming video processing. The mechanism to reduce inference frequency (only triggering high-level reasoning at key moments) effectively addresses the latency bottleneck in real-time applications.

3. The paper provides a wide range of baselines, comparing RVOS, Streaming Video, and MLLM methods. The experiments are well-organized, and the proposed metrics (like J_fg​ for foreground IoU) are reasonable for the defined problem setting.

Weaknesses:

1. Gap Between "Active Exploration" and "Passive Video". The paper positions itself as "Active Exploration," but the experimental setup is fundamentally strictly constrained by pre-recorded videos. This creates a significant gap between the claims and the evaluation. Firstly, I think it is lack of Agency: In a real "active" setting, an agent’s action changes the observation. Here, the model must predict an action that matches the camera operator's past decision. If the model decides to "turn left" but the video moves "forward," the evaluation treats this as a mismatch or tries to ignore it, but the visual feedback loop is broken. This setup resembles "Streaming Action Anticipation + Segmentation" rather than true active exploration.
Secondly, for a task emphasizing "exploration" and "planning," a 3D simulation environment (like Habitat-Lab or AI2-THOR) would be the standard and appropriate testbed. Evaluating navigation policies on offline videos limits the generalization of the findings to real-world embodied agents. The method essentially learns to imitate the cameraman's trajectory rather than learning a robust exploration policy.

2. The paper argues that planning and segmentation are tightly coupled. However, in the inference phase on a fixed video stream, the "planning" output (actions) does not influence the visual input of the next frame.
It raises the question: Is the segmentation performance improvement coming from "active planning," or simply from the powerful backbone (InternVL + SAM2) and the additional supervision from reasoning traces? The task could theoretically be decoupled: a pure segmentation model (using SAM2) might perform equally well if provided with the same video stream, rendering the "navigation/action" head trivial or redundant during inference.

---

> ### Author Rebuttal · Authors · 2026-03-31
>
> ## Rebuttal to Reviewer *XzAR*
> We thank the reviewer for the positive feedback and helpful suggestions, and address the remaining concerns below.
>
> ---
> **W1: The offline proxy evaluation setting of RVAS dataset vs. real closed-loop exploration.**
>
> We respect the reviewer's concern regarding the gap between proxy evaluation and real-world deployment. We would like to clarify two points.
>
> **(i) Real closed-loop benchmark is very hard.** A key design goal of RVAS is to study active exploration across diverse, realistic conditions, which remains underexplored in VLM and embodied AI fields. Our dataset spans 5 different scene types, open-set actions, and complex environments with dynamic objects and interactive elements. Existing 3D simulators (e.g., Habitat, AI2-THOR) are largely restricted to indoor navigation with a fixed action vocabulary and static scenes, making it hard to build a public closed-loop benchmark that faithfully supports these dimensions.
>
> **(ii) RVAS is valuable.** Importantly, RVAS is not intended to replace full embodied benchmarks, but to provide a **public and reproducible testbed** for studying goal-conditioned decision-making under streaming observations. Compared to closed-loop environments, which are often platform-dependent and difficult to standardize, our proxy setting enables controlled comparison across methods and isolates the core temporally stable, goal-oriented planning, and perception capabilities coupled with the exploration process. As discussed in Sec. 5.1, robust immediate behavior under this proxy setting is an essential prerequisite for successful active exploration. A model that fails in this setting is therefore very unlikely to perform well in closed-loop deployment. We further discuss which conclusions are and are not supported by the current offline evaluation in our response to reviewer **HAQy (Q1)**.
>
> To mitigate the limitations of pre-recorded trajectories, we adopt agreement-based filtering for val-set and preserve alternative valid actions during annotation. See Appendix A.2 and our response to reviewer **7ycJ (W1)** for details. Besides, an additional small-scale pilot study in a real-world setup is shown in the response to reviewer **1xBe (W1)**. The result reveals a consistent performance ranking with our offline benchmark.
>
> ---
> **W2 & Q2: The coupling between planning and segmentation under a fixed video stream.**
>
> The reviewer raises an important question regarding whether the performance gains are truly attributed to the action planning module, given that the video stream remains fixed during inference.
>
> We clarify that in LESA, the improvement is not solely from the backbone. Action prediction is not merely an auxiliary head, but an *internal conditioning signal* that directly affects segmentation. The predicted action calibrates $f_{ref}$ and updates the hidden state $h_t$ and memory $M_t$ via gated cross-attention, thereby re-conditioning attention to task-relevant cues in subsequent frames. As a result, the action head enables exploration-aware triggering, organizes historical information, and guides visual attention toward task-relevant cues.
>
> This effect is also supported empirically. In Table 5, removing action-conditioned history degrades performance (J&F: 57.4% → 54.8% / 56.2%; $J_{fg}$: 21.8% → 16.3% / 20.6%). In Table 6, removing action-guided gating further reduces performance (down to 46.2% J&F and 12.9% $J_{fg}$). Following the reviewer’s suggestion, we additionally train a variant without the action head; the results are shown below.
> |Method|Source|Desc|J&F|$J_{fg}$|
> |-|-|-|-|-|
> |I|Newly added|**NO ACTION HEAD**|53.1|10.4|
> |II|Tab. 5 (I)|no exploration-based history + no recent|54.8|16.3|
> |III|Tab. 5 (II)|no exploration-based history|56.2|20.6|
> |IV|Tab. 6 (I)|no action-gated + no query state|46.2|12.9|
> |V|Tab. 6 (II)|no action-gated + query state|53.2|19.7|
> |**LESA (Ours)**|Tab. 3|**WITH ACTION HEAD**|59.7|23.7|
>
> ---
> **Q1: Integration of LESA with next-generation segmentation models.**
> In our current implementation, SAM2 mainly serves as a mask decoder that translates the output token \<seg> from the multimodal model into pixel-level masks, similar to the design adopted in prior works such as VISA and Sa2VA. This is a mature and efficient design choice, which motivates our use of SAM2 in this work. Importantly, the mask generation process is not fundamentally tied to SAM2. In principle, LESA can be readily extended to integrate next-generation segmentation models (e.g., SAM3) by replacing the decoder that maps the segmentation token to the corresponding mask representation.
> That said, aligning large-model outputs with a new segmentation decoder typically requires substantial additional pretraining. Therefore, a fair comparison with alternative backbones is beyond the scope of the rebuttal, and we leave this to future work.

---

> > ### Author Rebuttal · Reviewer_XzAR · 2026-04-03
> >
> > I thank the authors for their detailed response and new experimental results during this short window.
> >
> > My primary concern was the fundamental gap between the offline proxy evaluation and true active exploration in a closed-loop setting. The additional real-world pilot study using the 6-DoF ROKAE robotic arm (provided in the response to Reviewer 1xBe) is a very strong addition. Demonstrating that the performance rankings from the offline benchmark translate consistently to a physical, closed-loop environment significantly alleviates my concerns regarding the practicality and agency of the proposed framework.
> >
> > Furthermore, I appreciate the newly added ablation experiment ("NO ACTION HEAD"). The noticeable drop in performance (e.g., J&F dropping from 59.7% to 53.1%) successfully answers my question about the coupling between planning and segmentation.
> >
> > I am raising my score to a Weak Accept. I strongly encourage the authors to include both the robotic pilot study and the new action-head ablation in the final manuscript or appendix, as they greatly strengthen the paper's claims.

---

> > > ### Author Response · Authors · 2026-04-03
> > >
> > > Dear Reviewer XzAR,
> > >
> > > Thank you sincerely for your time, your thoughtful feedback, and for acknowledging that our rebuttal has fully resolved your concerns. We are highly encouraged to hear that the newly added experiments successfully addressed your questions regarding the closed-loop evaluation and the coupling between planning and segmentation.
> > > As you strongly encouraged, we will include these parts in the final manuscript and appendix.
> > >
> > > We deeply appreciate your highly constructive comments, which have genuinely helped us strengthen the claims of our paper. Thank you again for your valuable time to support the work!
> > >
> > > Best regards,
> > > Authors of Paper #549

---

### Official Review · Reviewer_1xBe · 2026-03-08

**Soundness:** 3
**Presentation:** 1
**Significance:** 2
**Originality:** 2
**Overall Recommendation:** 4
**Confidence:** 3

**Summary:**

The paper introduces RVAS, a new task proposed by the authors that extends RVOS by adding actions. Considering scenarios where the referred object is fully visible in a givne video, the model has to determine its actions to locate the targeting object. Given the task is new, the authors create a dataset to support training. Additionally, the paper introduces LESA, which is a simple baseline that shows initial results on the proposed dataset.

**Compliance With Llm Reviewing Policy:**

Affirmed.

**Final Justification:**

The authors' rebuttal addressed my major concerns, thus maintaining my score.

**Key Questions For Authors:**

How much do the authors expect the results to change in a true closed-loop setting? Even a small pilot experiment would help here.

The dataset has only 911 videos and 5 scene types. Can the authors descirbe about on what kinds of generalization they expect from this scale?

My initial rating is between 3 and 4.

**Limitations:**

The paper does mention that the current datsaet & benchmakr is only a scalable proxy and not a real closed-loop embodied setting.

**Strengths And Weaknesses:**

The task is clear, and easy to understand; there are a number of extensions to existing tasks by adding actions, the extension of RVOS to RVAS is straightforward.

The soundness is somewhere between fair and good. The experiments are reasonably broad and design choices are validated through the ablations.
However, the evaluation makes the paper not a fully embodied setup; it evaluates on GT observation histories and assumes ideal execution, which makes it become a proxy benchmark, weakening the practicality. The authors provide some humna-alignment metrics, but it is unclear if a success in this benchmark would direclty transfer to real closed loop scenarios.

The presentation is fair to poor. As this paper is mainly a dataset paper, the dataset curation part is described is easy to follow. However, the method part, especially figure 1 and 3 are extremely dense and complicated that it is very hard to find it useful. In addition, I believe the authors should have attached sample data in the supplementary.

I believe the dataset contribution and significance is around fair. 911 videos is not that large, especially compared to several prior RVOS datasets listed in Table 1. The dataset is richer in annotation with some human reasoning annotations, and that is valuable. But in scale, it still feels more like a targeted benchmark than a major dataset resource for training. The fact that it only covers 5 scene types also makes it unclear how broadly it will generalize.

The originality is somewhere between fair to good as it proposes a new task and benchmark.

---

> ### Author Rebuttal · Authors · 2026-03-31
>
> ## Rebuttal to Reviewer *1xBe*
> We thank the reviewer for the detailed feedback and thoughtful comments. We appreciate that the reviewer finds RVAS task clear and straightforward to understand, and acknowledges the value of extending RVOS with action modeling.
>
> ---
> **W1 & Q1: The offline proxy evaluation setting of RVAS dataset vs. real closed-loop exploration.**
>
> Respect the concern regarding the gap between proxy evaluation and real-world deployment. We would like to clarify two points.
>
> **(i) Real closed-loop benchmark is very hard.** A key goal of RVAS is to study active exploration under diverse and realistic conditions, which remains underexplored in VLM and embodied AI. RVAS spans five scene types, open-set actions, and complex environments with dynamic objects and interactive elements. By contrast, existing 3D simulators (e.g., Habitat and AI2-THOR) are largely limited to indoor navigation with fixed action vocabularies and mostly static scenes, making it difficult to build a public closed-loop benchmark that faithfully covers these dimensions.
>
> **(ii) RVAS is valuable.** Importantly, RVAS is not intended to replace full embodied benchmarks, but to provide a **public and reproducible testbed** for studying goal-conditioned decision-making under streaming observations. Compared to closed-loop environments, which are often platform-dependent and difficult to standardize, our proxy setting enables controlled comparison across methods and isolates the core temporally stable, goal-oriented planning, and perception capabilities coupled with the exploration process. As discussed in Sec. 5.1, robust immediate behavior under this proxy setting is an essential prerequisite for successful active exploration. A model that fails in this setting is therefore very unlikely to perform well in closed-loop deployment. We further discuss which conclusions are and are not supported by the current offline evaluation in our response to reviewer **HAQy (Q1)**.
>
> To mitigate the limitations of pre-recorded trajectories, we adopt agreement-based filtering for val-set and preserve alternative valid actions during annotation. See Appendix A.2 and our response to reviewer **7ycJ (W1)** for details.
>
> (To other reviewers: pilot experiment addressed here)
>
> To further assess the practical relevance of our proxy evaluation, we conducted a small-scale real-world pilot study. We evaluated Sa2VA, InternVL3, and LESA using a 6-DoF ROKAE robotic arm with an omnidirectional camera platform, over 10 runs on each of 3 desktop scenes. The results are:
> |Model|SR|
> |-|-|
> |Sa2VA|23.3%|
> |InternVL3|36.6%|
> |LESA|43.3%|
>
> ---
> **W2: The presentation, and limited readability of Figures 1 and 3.**
>
> We will revise Figures 1 and 3 to reduce visual complexity and better highlight the core workflow and key components. We will also improve the presentation of the Method section for readability, and refer the reviewer to **HAQy (Minor)** for details.
>
> ---
> **W3: Lack of sample data in the supplementary.**
>
> We provide sample-data visualizations at https://anonymous.4open.science/r/ICML2026-rebuttal-549/data_samples.svg, including annotated frames, action sequences, and reasoning traces. These examples will be included in the supplementary material in the revised version.
>
> ---
> **W4: The scale of videos in the dataset and the scope of scene categories.**
>
> Thanks for the insightful comments on dataset scale and generalization.
>
> Regarding scene diversity, the five scene categories in RVAS denote high-level environments—desktop, single-room, multi-room, outdoor, and robotics—rather than a fine-grained enumeration of scenes. Each category further contains diverse layouts and concrete scenarios (e.g., bedroom, school, street, and desktop activity scenes). Compared with existing anticipation, navigation, and VLM datasets, RVAS covers a broader range of high-level settings. While it contains fewer total scenes than very large-scale segmentation datasets such as SA-V, RVAS is designed specifically for egocentric, exploration-based settings. We therefore believe that these five high-level categories provide sufficient coverage for the target task.
>
> Considering that RVAS provides longer videos on average (**∼21s**) than Ref-Youtube-VOS (∼5.5s) and MeViS (∼13s), together with rich annotations including **4,633 expressions, 18.6K annotated actions, and detailed reasoning traces**, we believe the current dataset scale is sufficient for model training. Under our action-level training strategy, RVAS offers adequate supervision to effectively fine-tune multimodal models at the 7B–8B scale.
>
> Overall, the combination of long-horizon videos, dense action annotations, and structured reasoning traces makes RVAS a rich and comprehensive dataset for the proposed task, allowing the trained model to generalize to (i) different scenes and layouts within each category, (ii) diverse action sequences and exploration strategies, and (iii) varying, diverse types of language instructions.

---

> > ### Author Rebuttal · Reviewer_1xBe · 2026-04-02
> >
> > The authors addressed most of my concerns, thus maintaining my score.

---

> > > ### Author Response · Authors · 2026-04-03
> > >
> > > Dear Reviewer 1xBe,
> > >
> > > Thank you sincerely for your time and for acknowledging to the rebuttal. We are very glad to hear that our clarifications were helpful. As promised, we will carefully incorporate the results, the dataset samples, and the revised Figures 1 and 3 into the final version of our paper to further improve its clarity and completeness.
> > > We deeply appreciate your constructive feedback and continued support for the work!
> > >
> > > Best regards,
> > > Authors of Paper #549

---

### Official Review · Reviewer_7ycJ · 2026-03-10

**Soundness:** 3
**Presentation:** 2
**Significance:** 3
**Originality:** 3
**Overall Recommendation:** 5
**Confidence:** 4

**Summary:**

Motivated by the limitations of existing Referring Video Object Segmentation (RVOS) research—such as the offline nature of the segmentation process, the assumption that the target object is always visible, and the difficulty of deploying Vision Object Navigation (VON) in real-world scenarios—this paper introduces a new task named Referring Video Active Exploration and Segmentation (RVAS). The proposed task requires the model to perform search and action planning before the target is observed. To support this task, the authors construct a dedicated dataset and evaluation metrics, and conduct extensive evaluations of existing methods. In addition, the authors design an efficient streaming framework tailored to this task, which achieves promising results.

**Compliance With Llm Reviewing Policy:**

Affirmed.

**Final Justification:**

There remains a gap between the current offline agent setup and true closed-loop active exploration, yet the authors have provided mitigations to the best of their ability. As an improvement over prior RVOS task, I believe this work is valuable and desirable to the research community, and I therefore decide to raise my score.

**Key Questions For Authors:**

See weeknesses

**Limitations:**

yes

**Strengths And Weaknesses:**

Strengths:
1. The paper presents a relatively complete piece of work with a convincing motivation. It includes the proposal of a new task, the construction of a dataset and evaluation metrics, and the development of a targeted technical framework (LESA).
2. The newly proposed task is meaningful from a research perspective. It extends traditional passive Referring Video Object Segmentation (RVOS) to an active exploration + segmentation paradigm, namely RVAS, which can also serve as a fine-grained evaluation benchmark for streaming video models and embodied intelligence.
3. The design of the LESA framework is reasonable and convincing at least at the level of its basic idea. In particular, selecting only key information for high-level reasoning is an effective strategy to mitigate the high computational cost of streaming video processing.

Weaknesses:
1. The evaluation is based on pre-recorded videos and cannot be validated in a closed-loop manner on real embodied intelligence platforms. While we understand that building such simulations can be extremely costly, the gap between simulation and real-world deployment inevitably limits the scope of the study. Considering this point, the progressiveness of this work is relatively limited.
2. As mentioned in W1, the predefined “ideal” decision trajectory imposes a rather strict evaluation criterion on the model. In practice, reaching a target often involves multiple practicable paths, which differs significantly from real-world scenarios. By the way, when the evaluated model outputs an action that deviates from the predefined trajectory at the current state, how do the authors handle this situation?
3. In terms of presentation, Figure 1 and Figure 3 contain overly dense and intertwined information, which increases the difficulty of reading and understanding them.

---

> ### Author Rebuttal · Authors · 2026-03-31
>
> ## Rebuttal to Reviewer *7ycJ*
> We thank the reviewer for the constructive feedback and insightful comments. We are encouraged that the reviewer finds the motivation of this work convincing and appreciates the completeness of our contribution, including the new task, the dataset and evaluation, and the LESA framework.
>
> ---
> **W1 & W2: The offline proxy evaluation setting of RVAS dataset vs. real closed-loop exploration.**
>
> We respect the reviewer's concern regarding the gap between proxy evaluation and real-world deployment. We would like to clarify three points.
>
> **(i) Real closed-loop benchmark is very hard.** A key design goal of RVAS is to study active exploration across diverse, realistic conditions, which remains underexplored in VLM and embodied AI fields. Our dataset spans 5 different scene types, open-set actions, and complex environments with dynamic objects and interactive elements. Existing 3D simulators (e.g., Habitat, AI2-THOR) are largely restricted to indoor navigation with a fixed action vocabulary and static scenes, making it hard to build a public closed-loop benchmark that faithfully supports these dimensions.
>
> **(ii) RVAS is valuable.** Importantly, RVAS is not intended to replace full embodied benchmarks, but to provide a **public and reproducible testbed** for studying goal-conditioned decision-making under streaming observations. Compared to closed-loop environments, which are often platform-dependent and difficult to standardize, our proxy setting enables controlled comparison across methods and isolates the core temporally stable, goal-oriented planning, and perception capabilities coupled with the exploration process. As discussed in Sec. 5.1, robust immediate behavior under this proxy setting is an essential prerequisite for successful active exploration. A model that fails in this setting is therefore very unlikely to perform well in closed-loop deployment. We further discuss which conclusions are and are not supported by the current offline evaluation in our response to reviewer **HAQy (Q1)**.
>
> **(iii) The progressiveness of RVAS is also significant.** We respectfully note that RVAS is not a minor extension of RVOS, but introduces a fundamentally new problem setting: models must reason about **where to look before the target is visible**, rather than segmenting already-observed objects. This shifts the focus from passive perception to **goal-driven exploration under partial observability**, which is not addressed by prior RVOS benchmarks.
>
> (Addressing Reviewers **HAQy**, **1xBe**, and **XzAR**: solutions for multiple practical paths are addressed here)
>
> The reviewer also raised the concern about possible practicable paths that deviate from the predefined trajectory.
> Regarding these inherent limitations, we adopt two strategies detailed in Appendix A.2 to partially alleviate the problem: (i) When building the validation set, we primarily retain samples with majority agreement on all of the decision points and actions in the video, which reduces ambiguity and retains samples that demonstrate demonstrably superior exploration strategies. (ii) we preserve and manually add *alternative valid actions* at each decision point (illustrated in Fig. 8) to avoid penalizing semantically reasonable but trajectory-divergent predictions. Some other samples are shown here: https://anonymous.4open.science/r/ICML2026-rebuttal-549/annotations.svg. Besides, an additional small-scale pilot study in a real-world setup is shown in the response to **1xBe (W1)**. The result reveals a consistent performance ranking with our offline benchmark.
>
> ---
>
> **W3: The limited readability of Figures 1 and 3.**
>
> We thank the reviewer for the helpful feedback. We will simplify Figures 1 and 3 to reduce visual complexity and improve readability. In particular, Figure 3 will be redesigned with a clearer and more modular presentation, following the refined structure in our response to reviewer **HAQy (Minor)**. Figure 1 will be improved similarly to better convey the core idea of the task.

---

> > ### Author Rebuttal · Reviewer_7ycJ · 2026-04-02
> >
> > We appreciate the detailed response. The discrepancy between offline evaluation and real closed-loop settings is recognized as a common concern among reviewers. We acknowledge the improvements made in this paper compared with RVOS, as well as the two mitigation strategies proposed by the authors. Nevertheless, We remain conservative regarding the overall advancement of this work due to this factual gap. In addition, We recommend the authors to provide the revised versions of Figure 1 and Figure 3.

---

> > > ### Author Response · Authors · 2026-04-06
> > >
> > > Dear Reviewer 7ycJ,
> > >
> > > Thank you sincerely for your time and for your detailed and thoughtful feedback. We truly appreciate your recognition of our improvements over prior RVOS task, and mitigation strategies to address the discrepancy between offline evaluation and real closed-loop settings.
> > >
> > > Regarding your suggestion, we have provided anonymized revised versions of Figure 1 (https://anonymous.4open.science/r/ICML2026-rebuttal-549/Figure_1_revised.pdf) and Figure 3 (https://anonymous.4open.science/r/ICML2026-rebuttal-549/Figure_3_revised.pdf) for your reference. If you have any other suggestions or expectations, as well as valuable perspective to the other respects of the work, please feel free to add comments at any time. We would be more than happy to further improve them based on your guidance and discuss your concerns.
> > >
> > > Thank you again for your constructive feedback and for your support of our work.
> > >
> > > Best regards,
> > > Authors of Paper #549

---

### Official Review · Reviewer_HAQy · 2026-03-12

**Soundness:** 3
**Presentation:** 3
**Significance:** 3
**Originality:** 4
**Overall Recommendation:** 4
**Confidence:** 3

**Summary:**

This paper proposes RVAS, a new task for referring video understanding. Different from normal referring video segmentation, the target may be not visible at the beginning, so the model needs to explore first and then segment the target after finding it. The paper also builds a new dataset with action annotations and reasoning traces, and proposes a streaming baseline LESA. The idea of using a light controller to decide when to call a heavy MLLM is practical. Experiments show good speed and competitive performance.

**Compliance With Llm Reviewing Policy:**

Affirmed.

**Final Justification:**

The rebuttal is helpful and clarifies several points, especially the intended scope of the offline benchmark, the role of the main modules in LESA, and the motivation of the current action evaluation protocol. However, my main concerns are still only partially addressed: the gap between the current offline proxy setting and real closed-loop active exploration still remains, and the possible uncertainty of LLM-judge / embedding-based action evaluation is more clarified than fully resolved. Therefore, the rebuttal does not change my overall assessment, and I maintain my original score.

**Key Questions For Authors:**

- Can the authors clarify more clearly what kinds of conclusions this offline evaluation can support, and what kinds it cannot support?

- Can the authors explain how they reduce possible bias from the LLM judge and embedding-based action matching?

- Can the authors clarify more clearly the role of each key module in LESA, especially the controller and the role-structured MLLM part?

**Limitations:**

The main limitation is that the benchmark is still based on proxy evaluation, so it cannot fully reflect real closed-loop exploration. Another limitation is that the action metric depends on external judge models, which may introduce some bias. These points should be stated more directly.

**Strengths And Weaknesses:**

Strengths

- The task setting is new and meaningful, and closer to real use cases.
- The paper provides task, dataset, metric, and baseline together, which is valuable.
- The method design is practical, especially the streaming control part.

Weaknesses

- Major 1: The evaluation is still an offline proxy setting, not real closed-loop exploration. The paper should make this limitation more clear, and be more careful when making claims about active exploration ability in real scenarios.

- Major 2: The action evaluation relies a lot on LLM judge and embedding-based matching. The paper has some explanation for this choice, but the possible bias and instability of such evaluation should be discussed more clearly.

- Minor 1: Some important method components are introduced, but their role in the whole framework is not always explained very clearly. So the contribution of each part is a little hard to follow.

- Minor 2: The writing is generally okay, but some parts in method and evaluation are still a bit dense. It would be better if the paper explains the protocol and metrics in a more direct way.

---

> ### Author Rebuttal · Authors · 2026-03-31
>
> ## Rebuttal to Reviewer *HAQy*
> We thank the reviewer for the positive feedback and valuable suggestions. We are pleased that the reviewer recognizes the **novelty and practical significance** of the proposed RVAS task, as well as **the value of presenting the task, dataset, metrics, and baseline** in a unified framework.
>
> ---
> **Major W1: The offline proxy evaluation setting of RVAS dataset vs. real closed-loop exploration.**
>
> We thank the reviewer for this important point. Due to the rebuttal length limit, we provide a detailed discussion of this limitation and its implications in our response to reviewer **7ycJ (W1)**, and a small-scale real-world pilot study is provided in the response to reviewer **1xBe (W1)**. In brief, real **closed-loop** environment is hard to build and **standardize**, and the RVAS proxy testbed is valuable.
>
> ---
> **Major W2 & Q2: The reliance of LLM judge and embedding-based matching for action evaluation.**
>
> We agree that LLM-based evaluation may introduce bias, and we take several steps to mitigate this.
>
> First, as discussed in Sec. 5.1, LLM-based evaluation is necessary due to the **open-set nature of actions and multiple valid annotations**, where traditional exact-match or n-gram metrics fail to capture semantic equivalence. Therefore, we adopt instruction-aware embedding similarity and a structured LLM judge for discrete matching under task-aware prompts.
>
> To improve robustness, the judge is calibrated with **carefully designed prompts and in-context examples**, including challenging cases. We enforce a deterministic setup, with an open-source backbone, temperature = 0, and structured outputs, to ensure reproducibility.
>
> We further validate reliability via **human alignment experiments** (Table 2), achieving high agreement (Cohen’s κ = 75.1%), and incorporate **alternative valid actions** (Appendix A.2) to reduce sensitivity to surface variations. Some additional analysis on hard cases (see https://anonymous.4open.science/r/ICML2026-rebuttal-549/llm_judge.svg) shows the judge remains largely consistent with human reasoning. We will clarify these points in the revision.
>
> ---
> **Minor W1, W2 & Q3: Unclear writing, especially in LESA method and evaluation sections.**
>
> (Addressing Reviewers **7ycJ** and **1xBe**: presentation-related questions are addressed here)
>
> We thank the reviewer for the helpful suggestion and will improve clarity in both the method and evaluation sections.
>
> In LESA, each module serves a distinct role:
>
> - **State Controller**: efficiently processes streaming observations and determines *when* high-level reasoning is needed. It maintains a compact hidden state with a **hierarchical memory design**, which preserves fine-grained recent observations while compressing long-term history under a fixed token budget. A **gated update mechanism** further conditions the state on the ongoing exploration process, ensuring sensitivity to action transitions and task progression.
>
> - **Role-Structured MLLM**: determines *what to do*, i.e., generating both action plans and segmentation outputs within a unified reasoning space via role switching.
>
> - **Exploration-Activated Calibration**: aligns perception with the current goal and predicted action, ensuring that subsequent attention focuses on task-relevant cues.
>
> We will revise the method section and simplify Figures 1 and 3 to better highlight these roles and reduce visual complexity.
>
> For evaluation, we will streamline the description of the protocol and metrics to make them more direct and easier to follow, including the embedding-based similarity metric to evaluate the overall alignment, the Acc-All and Acc-Any metrics to evaluate temporal accuracy and stability of predicted actions, and $J_{fg}$ to penalize the empty masks in long-horizon explorations.
>
> ---
> **Q1: The conclusions that can be supported and cannot be supported by current offline evaluation.**
>
> (Addressing Reviewers **7ycJ**, **1xBe**, and **XzAR**: conclusions supported and cannot be supported in the proxy setting are addressed here.)
>
> Our offline evaluation **can** reliably assess whether a model produces stable, temporally consistent, and goal-conditioned planning signals under streaming observations, as well as its ability of reliable segmentation based on long-horizon exploration. These capacities are the necessary prerequisites for any deployable active exploration agent. The results show LESA achieves ~10× lower latency with superior plan–segment consistency over prior baselines.
>
> However, the offline evaluation **cannot** characterize: (1) end-to-end exploration efficiency under real actuation (e.g., varying robot embodiments or execution delays), and (2) performance ceilings on specific hardware platforms. Closed-loop evaluation with physical actuators remains important future work, as noted in our Impact Statement.

---

> > ### Author Rebuttal · Reviewer_HAQy · 2026-04-04
> >
> > The rebuttal addressed part of my concerns, but my main concern is still not fully resolved. Therefore, I maintain my score.

---

> > > ### Author Response · Authors · 2026-04-04
> > >
> > > Dear Reviewer HAQy,
> > >
> > > Thank you very much for your time and for carefully reviewing our rebuttal. We truly appreciate your thoughtful feedback and are encouraged that our responses have addressed part of your concerns. We will incorporate the corresponding parts in the revisions.
> > >
> > > We warmly welcome any follow-up questions or further discussion, and we would be happy to continue addressing your concerns in more detail. Please feel free to let us know if there are specific problems you would like us to clarify more.
> > >
> > > Thank you again for your time and support to the work!
> > >
> > > Best regards,
> > > Authors of Paper #549

---

### Decision · Program_Chairs · 2026-04-30

**Decision:**

Accept (regular)

**Comment:**

The paper introduces Referring Video Active Exploration and Segmentation (RVAS), advancing the passive RVOS task by requiring models to perform search and action planning before segmenting a target. The authors contribute a dataset of 911 videos with reasoning traces and propose a streaming baseline, LESA.

Reviewers recognized the task as highly meaningful and found the LESA framework to be logically designed for streaming video processing. However, a unanimous criticism across all reviewers was the evaluation's reliance on pre-recorded videos (a proxy setting) rather than true closed-loop active exploration.

During the rebuttal, the authors defended the necessity of the proxy benchmark for reproducible evaluation and provided a small-scale real-world robotic pilot study, which successfully demonstrated that offline benchmark rankings translate consistently to a physical environment. While the offline evaluation remains an inherent limitation of this work, the introduction of the RVAS task, the comprehensively annotated dataset, and the streaming baseline offer a highly valuable "stepping stone" for future research in embodied AI and video modeling. Therefore, the paper is recommended for a Weak Accept.